# TPC2 promotes choroidal angiogenesis and inflammation in a mouse model of neovascular age-related macular degeneration

Yanfen Li[1], Christian Schön[1], Cheng-Chang Chen[1,2], Zhuo Yang[1], Raffael Liegl[3], Elisa Murenu[1], Benedikt Schworm[3], Norbert Klugbauer[4], Christian Grimm[6], Christian Wahl-Schott[5], Stylianos Michalakis[1,3], Martin Biel[1]

**Age-related macular degeneration (AMD) is the most common cause of blindness among the elderly and can be classified either as dry or as neovascular (or wet). Neovascular AMD is characterized by a strong immune response and the inadequate release of cytokines triggering angiogenesis and induction of photoreceptor death. The pathomechanisms of AMD are only partly understood. Here, we identify the endolysosomal two-pore cation channel TPC2 as a key factor of neovascularization and immune activation in the laser-induced choroidal neovascularization (CNV) mouse model of AMD. Block of TPC2 reduced retinal VEGFA and IL-1β levels and diminished neovascularization and immune activation. Mechanistically, TPC2 mediates cationic currents in endolysosomal organelles of immune cells and lack of TPC2 leads to reduced IL-1β levels in areas of choroidal neovascularization due to endolysosomal trapping. Taken together, our study identifies TPC2 as a promising novel therapeutic target for the treatment of AMD.**

## Introduction

Age-related macular degeneration (AMD) is the major cause of blindness among the elderly ([1]). The estimated prevalence of the disease is expected to be close to 300 million worldwide by 2040, posing a significant global economic and clinical burden ([2]). AMD is characterized by dysfunction and degeneration of photoreceptors and the retinal pigmented epithelium (RPE) and vascular changes. Early clinical signs include the formation of yellowish white deposits of cellular debris (the so-called Drusen) between the RPE and Bruch's membrane and morphological changes of the RPE itself. Late-stage AMD occurs in two main clinical forms, the dry form also known as geographic atrophy and the wet or neovascular form. Dry AMD is characterized by progressive atrophy of the outer retinal layers and the RPE, whereas the wet form is characterized by the invasion of leaky blood vessels from the choroid (choroidal neovascularization, CNV) into the retina and local inflammation that together lead to severe damage of retinal photoreceptors and impaired vision ([3]). Both types of AMD are pathophysiologically connected and dry AMD has been considered to be the precursor of neovascular AMD. So far, the pathogenesis of neovascular AMD is only partly understood ([4]). However, there is consensus that a variety of factors, including age, genetic polymorphisms, nicotine abuse, environmental risk factors and systemic health, play a role in onset, severity and progression of the disease ([5]). There is also growing evidence that immune activation ([6], [7], [8]), in particular the activation of the inflammasome ([9]) and the release of proinflammatory cytokines (e.g., IL-1β and IL-18) by microglia and infiltrating macrophages ([10]) is crucially involved both in the initiation and the progression of neovascular AMD. Cells of the innate immune system (microglia and macrophages) promote the early development of CNV by secretion of proangiogenic growth factors (e.g., VEGFA) ([11]) and by triggering the release of such factors from other retinal cell types, among which the RPE is most prominent ([12]).

Two-pore channels (TPCs) are $Ca^{2+}$ and $Na^+$ permeable cation channels that are expressed in endolysosomal organelles of many tissues and cell types ([13], [14], [15]). Two members exist in mice and humans, namely, TPC1 and TPC2 ([16], [17], [18], [19]). TPCs have been shown to participate in the regulation of multiple endolysosomal trafficking pathways and are implicated in various pathological processes, such as cancer cell migration ([20]), as well as metabolic ([19], [21]) and infectious diseases ([22]). TPCs have also been implicated in immune signaling processes, including macrophage phagocytosis ([23]), mast cell degranulation ([24]) and exocytosis of lytic granules in CD8[+] T lymphocytes ([25]). Moreover, there is evidence for

[1]Department of Pharmacy, Ludwig-Maximilians-Universität München, München, Germany    [2]Department of Clinical Laboratory Sciences and Medical Biotechnology, College of Medicine, National Taiwan University, Taipei, Taiwan    [3]Department of Ophthalmology, University Hospital, LMU Munich, München, Germany    [4]Institute for Experimental and Clinical Pharmacology and Toxicology, Medical Faculty, Albert-Ludwigs-University, Freiburg, Germany    [5]Institute for Neurophysiology, Hannover Medical School, Hannover, Germany    [6]Walther Straub Institute of Pharmacology and Toxicology, Ludwig-Maximilians-Universität München, München, Germany

Correspondence: biel@lmu.de; michalakis@lmu.de

a role of TPC2 in signaling pathways that mediate VEGFA-induced angiogenesis (26). Because dysregulation of immune cells and aberrant angiogenesis are major clinical hallmarks of neovascular AMD, we reasoned that TPC2 may be involved in this AMD pathology. Indeed, using the mouse model of laser-induced CNV in combination with genetic and pharmacological tools we discovered a major role of TPC2 in inflammatory and neoangiogenic processes linked to AMD. Genetic deletion or acute pharmacological block of the channel prevented release of key pathogenic cytokines and protected mice from developing choroidal neovascularization after laser damage.

# Results

### Inhibition of TPC2 activity protects from excessive choroidal neovascularization in a mouse laser photocoagulation model of wet AMD

To explore the potential role of TPCs in CNV pathogenesis, we used a laser coagulation model that has been extensively used to study the exudative form of AMD in mice and other animals (27). In this CNV model, Bruch's membrane is damaged by focal laser photocoagulation, resulting in an inflammatory response and invasion of the retina by novel formed and leaky choroidal blood vessels that can be non-invasively monitored by fluorescein fundus angiography (FFA).

Fig 1A shows representative fundus images with laser burns on day 0 and FFA images obtained at 7 and 14 d after laser treatment from $Tpc2$-deficient ($Tpc2^{-/-}$) and a litter-matched wild-type (WT) mice. Four laser burns per eye were successfully induced in both WT and $Tpc2^{-/-}$ mice as indicated by Fig 1A (left panel). In WT mice, extended leakage areas at the positions of the four laser spots (marked with dotted lines in Fig 1A, upper panel) are clearly visible. By contrast, in the retina of $Tpc2^{-/-}$ mice the extent of the leakage areas was strongly reduced (Fig 1A, lower panel). The laser-induced CNV is a self-limiting model of neoangiogenesis and the lesion peaks at days 7–10 and regresses 2–3 wk later. We, therefore, first quantified the leakage area on day 7 and observed an about 50% reduction in $Tpc2^{-/-}$ versus WT control mice (Fig 1B; n = 7–9 eyes/group). At day 14, the extent of the leakage area started to decline, but the dampening effect of TPC2 deletion still persisted and remained at a similar level (Fig 1C). Because TPC1 is closely related to TPC2, we also analyzed $Tpc1^{-/-}$ mice but did not observe a difference compared with WT mice (Fig S1A and B). Therefore, we focused in the following on TPC2. To exclude that the differences observed in $Tpc2^{-/-}$ mice are due to a developmental effect caused by the deletion of TPC2 we tested the effect of the established TPC blockers tetrandrine (22) and Ned-19 (28) on CNV formation in WT mice. Each of the compounds was applied immediately before the laser treatment via single intravitreal injection with 1 µl of volume and the extent of CNV was evaluated after 7 and 14 d. Both tetrandrine (Fig 2A, C, and D and S2A) and Ned-19 (Figs 2B, E, and F and S2B) significantly inhibited the laser-induced neovascularization to an extent similar to the genetic deletion of TPC2.

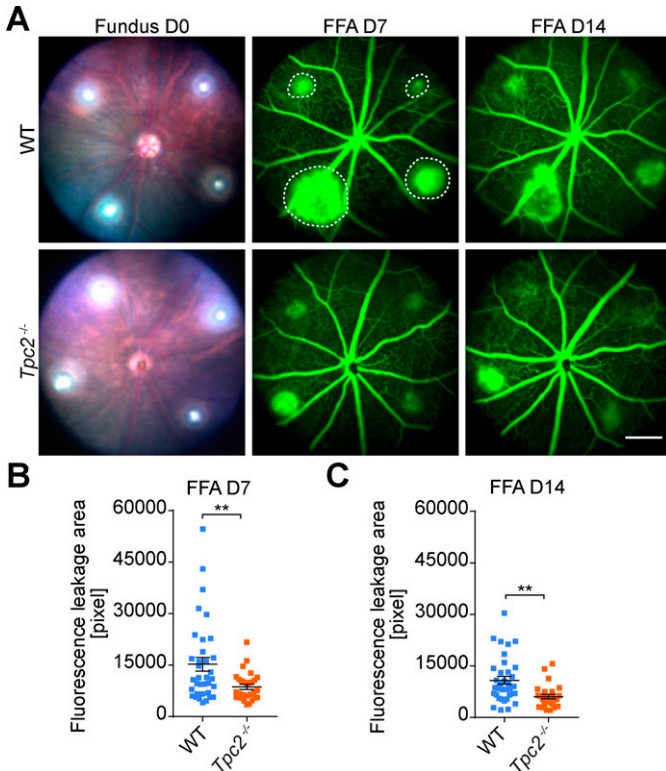

**Figure 1. Genetic deletion of two-pore channel-2 attenuates choroidal neovascularization (CNV) in a laser-induced mice model.**
**(A)** Representative fundus images on day 0 (D0) and fluorescein angiography (FFA) of subretinal lesion from 6- to 8-wk-old $Tpc2^{-/-}$ mice (A, lower panel), litter-matched WT mice (A, upper panel) 7 (D7) and 14 d (D14) after laser photocoagulation. Four laser-induced leakage areas are indicated by dotted line in (A). Scale bar = 300 µm. **(B, C)** Quantification of vascular leakage areas by analyzing pixel intensities after laser-induced damage for WT versus $Tpc2^{-/-}$ mice on day 7 (B) and day 14 (C). Each data point represents one lesion site on day 7 or day 14 after laser photocoagulation. Data are presented as mean ± SEM. Group size as n = 7 to 9 mice. Two-tailed $t$ test was used for statistical analysis. *$P$ < 0.05, **$P$ < 0.01, ***$P$ < 0.001.

### Block of TPC2 inhibits choroidal sprouting

To test whether TPC2 is directly involved in choroidal neovascularization we dissected pieces of the choroid (including attached RPE and sclera) from WT and $Tpc2^{-/-}$ mice, embedded the tissue in matrigel and cultured it in endothelial cell (EC) growth medium containing VEGFA to induce microvascular sprouting, which was visualized under a microscope and quantified using a computerized quantification method (SWIFT macro in ImageJ64 software) (29). After 4 d in culture, WT tissue revealed substantial outgrowth of newly formed sprouts (Fig 3A, upper row). In addition to ECs, choroidal sprouts contained pericytes (positive for chondroitin sulfate proteoglycan neuron-glial antigen 2, NG2), and fibroblasts (positive for vimentin) (Fig S4A and B). Although sprouting was also induced in tissue from $Tpc2^{-/-}$ mice, it was less pronounced than in tissue from WT mice (Fig 3A, lower row). Quantitative image analysis revealed that deletion of TPC2 caused an about 50% reduction of the sprouting area (Fig 3B; n = 8 pieces/group). Acute pharmacological blockade of TPC2 in WT tissue by

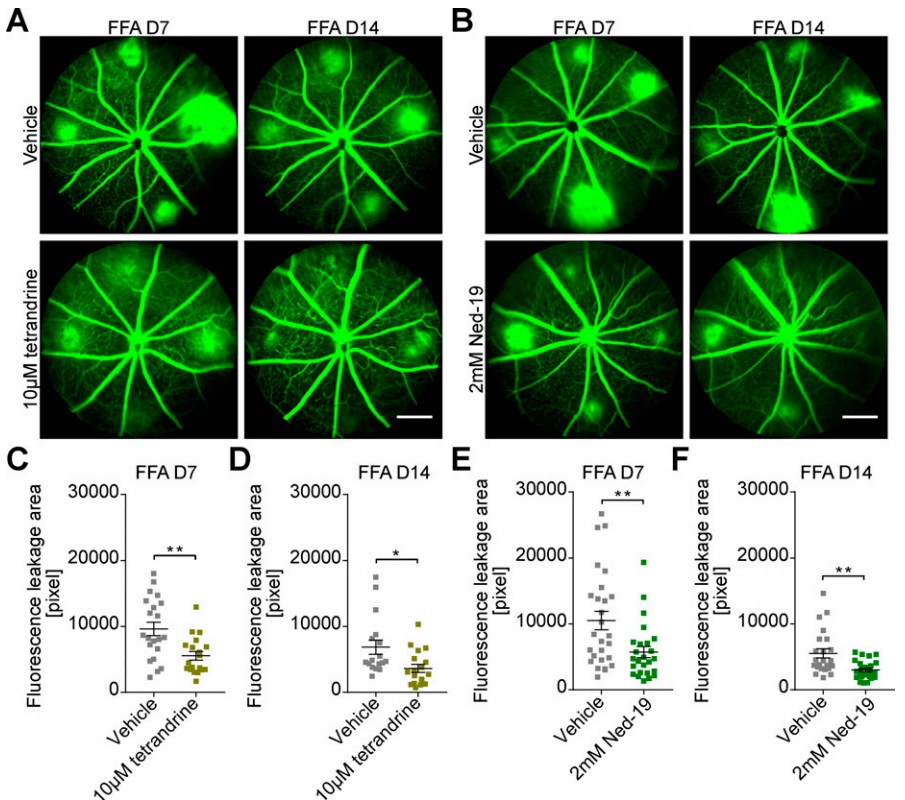

**Figure 2. Pharmacological block of two-pore channel-2 activity ameliorates choroidal neovascularization (CNV) in a laser-induced mice model.**
Representative FFA images of C57BL/6J WT mice (A, B) 7 and 14 d after laser photocoagulation. **(A, B)** FFA images of mice that were intravitreally injected into the right eyes with 1 μl of either 10 μM tetrandrine (A, lower panel) or 2 mM Ned-19 (B, lower panel) immediately before laser coagulation. **(A, B)** Upper panels in (A, B) show FFA images of vehicle-injected left eyes of the respective animals. **(A, B)** Scale bar in (A) and (B) = 300 $\mu m$. **(C, D, E, F)** Quantification of vascular leakage areas by analyzing pixel intensities after laser-induced damage for vehicle versus tetrandrine-treated (C, D) and vehicle versus Ned-19 treated WT mice (E, F). **(C, D, E, F)** Each data point represents one lesion site on day 7 or day 14 after laser photocoagulation. Data are presented as mean ± SEM. Group size as n = 6 in (C, D, E, F). Two-tailed $t$ test was used for statistical analysis. *$P$ < 0.05, **$P$ < 0.01, ***$P$ < 0.001.

either tetrandrine (Figs 3C and S3A) or Ned-19 (Figs 3D and S3B) also inhibited sprouting quantitatively and to a similar extent as the TPC2 knockout.

## TPC2 deficiency results in reduced infiltration of Iba-1+ cells and complement activation in CNV lesions

To further corroborate the in vivo data, RPE/choroid flat mounts from CNV-treated eyes were stained for the EC marker isolectin B4 (IB4) and the macrophage/microglia marker Iba-1 (Fig 4A, left panels). Quantification of the IB4-positive area confirmed the reduced laser-induced neovascularization in $Tpc2^{-/-}$ mice (Fig 4C). In the WT, the IB4 labeling also revealed a substantial amount of macrophage/microglia-like cells surrounding the lesions, which were also immunopositive for Iba-1 (Fig 4A). By contrast, the number of such Iba-1–positive cells in the lesion area of $Tpc2^{-/-}$ mice was significantly lower than in the WT controls (Fig 4A and D). Activation of the complement system plays a key role in the pathogenesis of neovascular AMD. In particular, assembly of the membrane attack complex (MAC) that triggers downstream activation of the Nod-like receptor protein 3 inflammasome is considered a major factor in the inflammatory reaction of AMD (30). In agreement with this model, a clear signal for the terminal complement complex C5b-9 that forms MAC could be detected in lesion areas of WT mice (Fig 4B, upper panel). In line with our previous findings, the C5b-9–positive signal was lower and more restricted in laser lesions of $Tpc2^{-/-}$ than in those of WT mice (Fig

4B lower panel and Fig 4E). Because VEGFA plays a key role in triggering neovascularization in AMD, we next determined the levels of VEGFA in retinas of WT and $Tpc2^{-/-}$ mice at 48 h after laser coagulation (Fig 4F). VEGFA levels were indeed also reduced in knockout mice compared with controls (93.14 ± 21.73 pg/mg in WT retina; 25.57 ± 5.738 pg/mg in $Tpc2^{-/-}$ retina, n = 5 to 6 eyes/group).

## TPC2 channels are functionally expressed in macrophages and microglia

The data collected so far suggest that block of TPC2 function dampens the inflammatory response and the extent of neo-angiogenesis in the mouse CNV model. In line with this, we found $Tpc2$ gene expression in RPE cells, peritoneal and BMDM, and at even higher levels in brain and retinal microglia (Fig 5A). To further corroborate this finding, we performed electrophysiological recordings in vacuolin-enlarged endolysomes of mouse macrophages and microglia. To specifically activate voltage-independent TPC2-like currents, low concentration of PI(3,5)P$_2$ (1 μM) was used in the presence of Na$^+$ and Ca$^{2+}$ as the major permeant ions (19, 31). TPC2-like currents were present in endolysosomes of BMDMs from WT mice (Fig 5B and D), whereas such currents were not detectable in $Tpc2^{-/-}$ BMDMs (Fig 5C and D). Similarly, TPC2-mediated currents were detected in brain microglia (Figs 5E and S6C and D) and peritoneal macrophages (PMs) (Figs 5F and S6A and B) from WT, but not from $Tpc2^{-/-}$ mice.

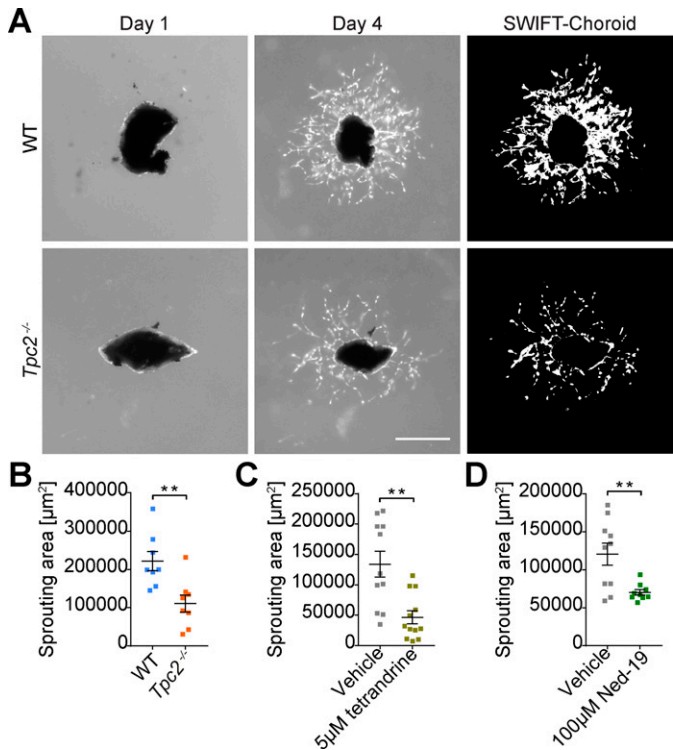

**Figure 3. Block of two-pore channel-2 inhibits choroidal sprouting.**
**(A)** Ex vivo choroidal sprouting assay of retinal tissue biopsies comprising retinal pigmented epithelium (RPE), choroid and sclera. Tissue was dissected from peripheral region of eyes of 21–25-d-old WT and $Tpc2^{-/-}$ mice, embedded in Matrigel, and cultured in medium. Microvascular sprouts were visualized using microscopy and processed and quantified with the SWIFT-Choroid macro in ImageJ64 software (National Institutes of Health). Representative images taken at day 1 and day 4 in culture (original and after SWIFT processing) are shown as indicated. Scale bar = 500 μm. **(B, C, D)** Quantification of the vascular sprouting area of WT versus $Tpc2^{-/-}$ mice (B), WT tissue in the absence or presence of 5 μM tetrandrine (C) or 100 μM Ned-19 (D) in the medium. Each data point represents one RPE/choroid/sclera piece on day 4. **(B, C, D)** Data are presented as mean ± SEM. n = 8 pieces/group in (B) and 9–12 pieces/group in (C, D). Two-tailed $t$ test, *$P$ < 0.05, **$P$ < 0.01, ***$P$ < 0.001.

## TPC2 deficiency impairs IL-1β secretion in mouse macrophages and microglia

After identifying TPC2 in immune cells, we set out to investigate how activity of this channel could be linked to the inflammatory response observed in the CNV model. A major proinflammatory factor that has been associated with the pathology of neovascular AMD is IL-1β (10). IL-1β is synthesized in microglia and macrophages as biologically inactive pro-IL-1β in response to various noxious stimuli (e.g., pathogens or cell damage) (10). After activation of the Nod-like receptor protein 3 inflammasome, pro-IL-1β is cleaved by the proinflammatory protease caspase-1 to mature IL-1β, which is subsequently secreted through different pathways, including secretory lysosomes (32). We therefore quantified global IL-1β levels in retinal homogenates from laser-photocoagulated WT and $Tpc2^{-/-}$ retina. Interestingly, we found significantly lower levels of IL-1β in the $Tpc2^{-/-}$ than in the WT retina (Fig 6A) (13.57 ± 1.232 pg/mg in WT retina; 9.449 ± 0.2109 pg/mg in $Tpc2^{-/-}$ retina, n = 6 eyes/group). To further investigate this, we immunolabeled retinal cross sections

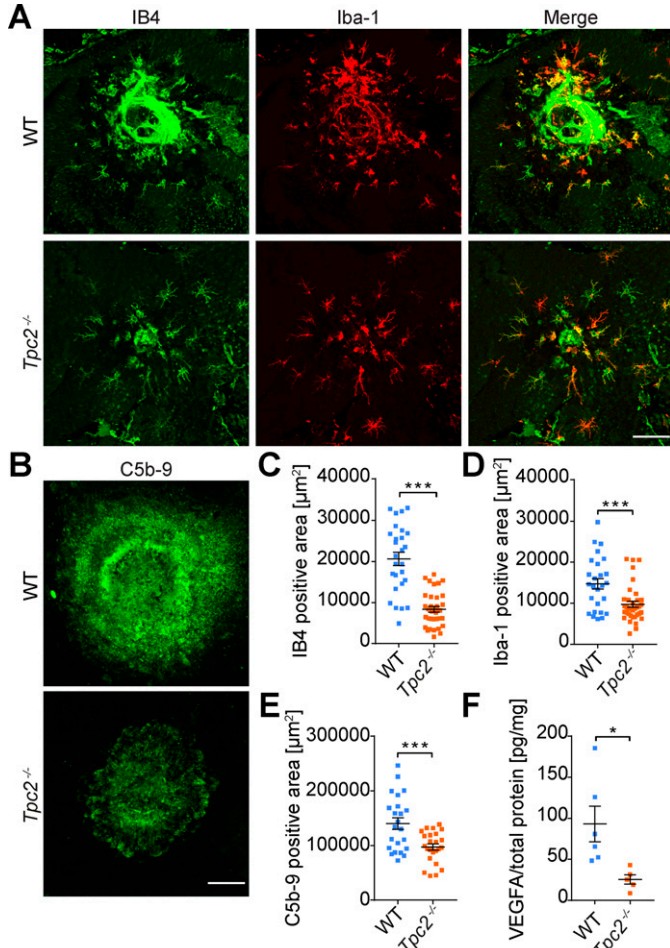

**Figure 4. $Tpc2^{-/-}$ mice display reduced infiltration of inflammatory cells and lower C5b-9 and VEGFA levels after laser damage.**
**(A)** Representative images of lesion sites in retinal pigmented epithelium/choroid flat mounts prepared from WT and $Tpc2^{-/-}$ mice 14 d after laser coagulation. Whole mounts were double immunostained with the endothelial cell marker isolectin B4 (green) and the microglia/macrophage marker Iba-1 (red). Scale bar = 100 μm. **(B)** Representatives images of C5b-9-stained (green) lesion area of WT and a $Tpc2^{-/-}$ mice. Scale bar = 75 μm. **(C, D, E)** Quantification of isolectin B4 (C), Iba-1 (D) and C5b-9 (E) stained area. Each dot represents one CNV lesion site. n = 24–36 lesions/group. Data are presented as mean ± SEM. Two-tailed $t$ test, ***$P$ < 0.001. **(F)** ELISA-based quantification of VEGFA levels in retinas from WT and $Tpc2^{-/-}$ mice on day 2 after laser photocoagulation. Each dot represents one eye. Data are presented as mean ± SEM, n = 6 eyes/group. Two-tailed $t$ test, **$P$ < 0.01, ***$P$ < 0.001.

for IL-1β and Iba-1. Irrespective of the genotype, little to no Iba-1+ or IL-1β signal was observed in non-lesioned parts of the retina (Fig S7A, upper panels). In contrast, within the lesion area of the WT retina, a high number of Iba-1+ cells and a clear IL-1β staining were observed in the subretinal areas (Fig S7A, lower panels). Although IL-1β and Iba-1 signals partially overlapped (indicated by arrowheads), most of the IL-1β signal distributed outside Iba-1+ cells, which probably reflected the presence of secreted IL-1β. The number of Iba-1–positive cells and the overall IL-1β positive area in the subretinal region was significantly smaller in $Tpc2^{-/-}$ than in WT retina (Fig S7B and C). However, some $Tpc2^{-/-}$ Iba-1+ cells displayed a strong IL-1β co-staining (Fig 6B, lower panels), suggesting

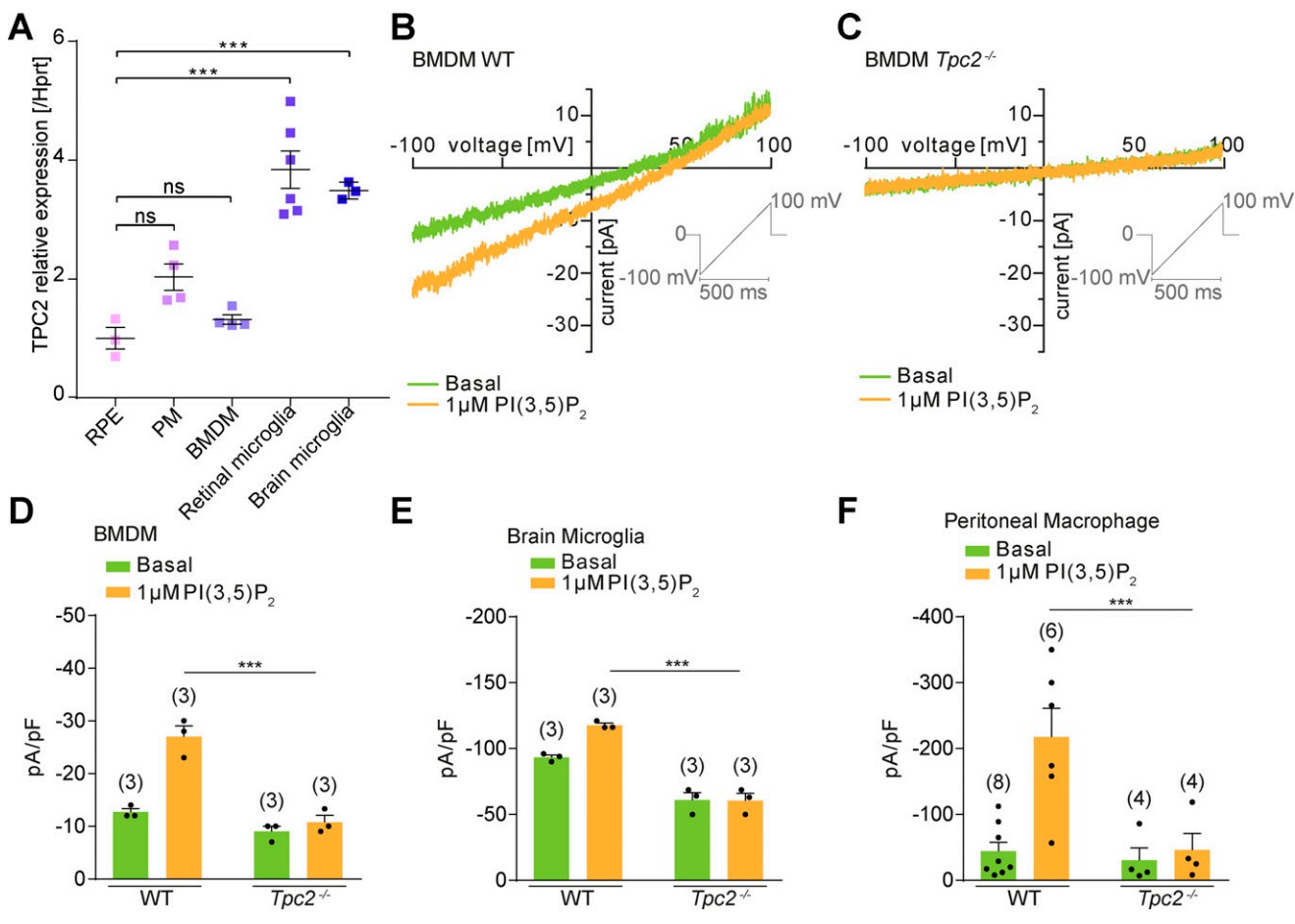

**Figure 5. Functional characterization of two-pore channel (TPC)-2 in macrophages and microglia.**
**(A)** qRT-PCR analysis of TPC2 expression in cultivated murine retinal pigmented epithelium, peritoneal macrophages, BMDM, and brain and retinal microglia. Relative TPC2 mRNA expression was normalized to hypoxanthine phosphoribosyl transferase (Hprt) mRNA expression. TPC2 mRNA expression level in retinal pigmented epithelium cells was set to 1. Three to six independent experiments were performed in duplicate. Data are presented as mean ± SEM. One-way ANOVA followed by Tukey's multiple comparisons test was used, ***$P < 0.001$, ns, no significant difference. **(B, C)** Representative current–voltage relationships recorded from vacuolin-enlarged endolysosomal vesicles isolated from cultured WT (B) or $Tpc2^{-/-}$ (C) BMDM. Currents were obtained before and after application of 1 μM phosphatidylinositol 3,5-bisphosphate (PI(3,5)P$_2$).**(B, C, D, E, F)** Statistical analysis of current densities recorded at −100 mV for currents as shown in (B) and (C) and for currents obtained under identical conditions from brain microglia (E) and peritoneal macrophage (F). **(D, E, F)** Data in (D, E, F) are presented as mean ± SEM. To test for statistical significance, two-way ANOVA was applied, ***$P < 0.001$.

an intracellular accumulation of IL-1β in $Tpc2^{-/-}$ cells. Fluorescence intensity quantification indeed confirmed an increase in IL-1β signal in $Tpc2^{-/-}$ Iba-1+ cells (Fig 6C). To further explore this finding, we analyzed IL-1β secretion in cultured BMDMs of WT and $Tpc2^{-/-}$ mice (Fig 7). Priming of BMDMs with LPSs led to a dramatic up-regulation of pro-IL-1β transcript (Fig S8) and protein (Fig 7A and B) in both WT and $Tpc2^{-/-}$ BMDMs. To quantify the amount of secreted IL-1β we first treated LPS-primed cells with ATP (33) to induce cleavage of pro-IL-1β and secretion of IL-1β. Subsequent quantification by ELISA revealed significantly lower levels of secreted IL-1β in $Tpc2^{-/-}$ than in WT BMDM cell culture supernatants (Fig 7C). These findings suggested that TPC2 is not essential for principal priming of BMDMs and expression of IL-1β, but is involved in the release of IL-1β. If this is the case, the expressed IL-1β should accumulate in intracellular compartments of $Tpc2^{-/-}$ cells. Indeed, isolated $Tpc2^{-/-}$ BMDMs challenged with LPS/ATP revealed a strong intracellular IL-1β signal that mainly co-localized with the lysosomal marker

LAMP-1 (Fig 8A, second row). Both, the IL-1β and LAMP-1 signal was significantly weaker in WT cells (Fig 8A, first row). Fluorescence intensity analysis of the staining showed an increase for $Tpc2^{-/-}$ cells (IL-1β: 63.27 ± 1.841; LAMP-1: 41.87 ± 1.595; n = 50) compared with WT cells of 51.17 ± 1.701 (n = 45) for IL-1β (Fig 8B) and of 35.3 ± 1.512 (n = 45) for LAMP-1 (Fig 8C). Increased intracellular accumulation of IL-1β in LAMP-1+ organelles was also observed in LPS/ATP-stimulated retinal microglia derived from $Tpc2^{-/-}$ mice (Fig 8D). Again, the overall signal intensity for IL-1β (49.35 ± 3.282 versus 27.12 ± 1.581, n = 45), and LAMP-1 (69.82 ± 4.139 versus 36.23 ± 3.269, n = 46) was higher in $Tpc2^{-/-}$ than in WT cells (Fig 8E and F).

## Discussion

Here, we identified the endolysosomal cation channel TPC2 as an important regulator of key processes involved in the pathology of

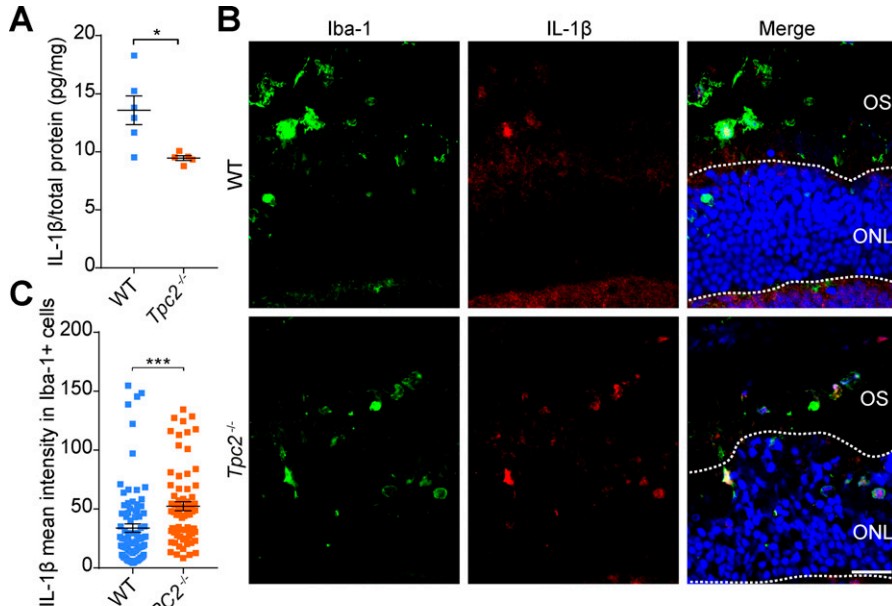

**Figure 6.** *Tpc2^{−/−}* **mice display reduced overall amounts of IL-1β in CNV areas and increased amounts of IL-1β in Iba-1–positive cells.**
**(A)** Quantification of IL-1β. Retinas were dissected and homogenized from WT and *Tpc2^{−/−}* mice on day 1 after laser photocoagulation. IL-1β level in the retina was determined by ELISA. A reduced IL-1β concentration was found in *Tpc2^{−/−}* retinas compared with WT control retinas. Each dot represents one eye. Data are presented as mean ± SEM, n = 6 eyes/group. Two-tailed *t* test, *$P < 0.05$. **(B)** Immunolabelled cross sections of retinas from WT and *Tpc2^{−/−}* mice on day 1 after laser treatment. Sections were co-stained for Iba-1 (left panels) and IL-1β (middle panels). The right panels show overlay of the two respective images. Scale bar = 20 *μm*. ONL, outer nuclear layer; OS, outer segment. **(C)** Quantification of IL-1β mean intensity in Iba1+ cells from WT and *Tpc2^{−/−}* retinas. Each dot represents one cell. n = 72–86 cells/group. Data are presented as mean ± SEM. Two-tailed *t* test, ***$P < 0.001$.

neovascular AMD. Although there is currently no model system available that covers all aspects of human AMD, the murine laser-induced CNV model used in our study is commonly used to study neoangiogenic and inflammatory components of the disease, in particular as a scar formation model it mimics the end stage of CNV (27). We found that mice in which *Tpc2* has been genetically deleted display significantly less neovascularization than WT mice at 7 and 14 d after laser damage. This effect was specific for *Tpc2* and did not occur in mice carrying a deletion of the homologous *Tpc1*. There was also no change in choroidal neovascularization in mice deficient for the lysosomal TRPML1 channel (not shown). Together, these findings indicate that the effects observed in *Tpc2^{−/−}* mice do no result from general lysosomal dysfunction but are rather reflecting the absence of a TPC2-specific function. Importantly, intravitreal application of one of the established TPC channel inhibitors, tetrandrine or Ned-19 also reduced the extent of laser-induced choroidal neovascularization in WT mice. This finding confirms that lack of TPC2 activity per se, rather than developmental or

compensatory effects induced by the TPC2 knockout, is responsible for the protection from choroidal neovascularization. Genetic deletion or pharmacological inhibition of TPC2 also impacted the blood vessel formation in response to VEGFA and other growth factors in the in vitro choroid sprouting assay. This suggests that the effect on neovascularization at least partially depends on VEGFA, which is in agreement with previous findings from Favia et al (26) who demonstrated that TPC2 provides a endolysosomal Ca^{2+} signal in response to activation of the VEGFA receptor type 2 (VEGFR2) that controls growth of vascular ECs. The vascular sprouts from the choroidal explant mainly consist of ECs surrounded by pericytes (29). The CD68-positive monocytes were also detected adjacent to planted choroidal tissue, but not at the tips of the sprouts (29). Although we have not specifically analyzed the corresponding signaling in ECs, it might be that the main effect of TPC2 in choroidal sprouting culture is attributable to its regulation in choroidal ECs as shown for vascularization of tumors (26). However, it might also involve additional effects of TPC2 in other cells such as pericytes or

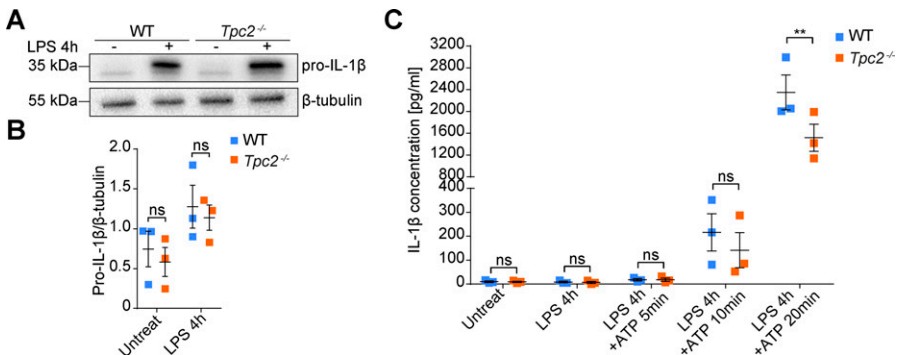

**Figure 7.** **Two-pore channel-2 is not crucial for IL-1β expression but is required for normal IL-1β secretion in macrophages.**
**(A)** Western blot analysis of cell lysates from cultured BMDMs of WT and *Tpc2^{−/−}* mice. The blot was probed for IL-1β and β-tubulin as loading control. In cells analyzed in lane 2 and 4, LPSs were added at 1 *μg*/ml to the medium 4 h before lysis to induce pro-IL-1β expression. **(B)** Densitometric quantification of pro-IL-1β expression. Three independent experiments were performed. Data are presented as mean ± SEM. Two-way ANOVA, ns, no significant difference. **(C)** Determination of secreted IL-1β. WT and *Tpc2^{−/−}* BMDMs were pretreated with or without 1 *μg*/ml LPS for 4 h to induce IL-1β expression, followed by stimulation with 5 mM ATP for 5, 10, and 20 min to

promote IL-1β maturation and secretion. Extracellular medium was collected and processed for ELISA analysis. Data are presented as mean ± SEM. Two-way ANOVA, *$P < 0.05$, **$P < 0.01$, ***$P < 0.001$, ns, no significant difference.

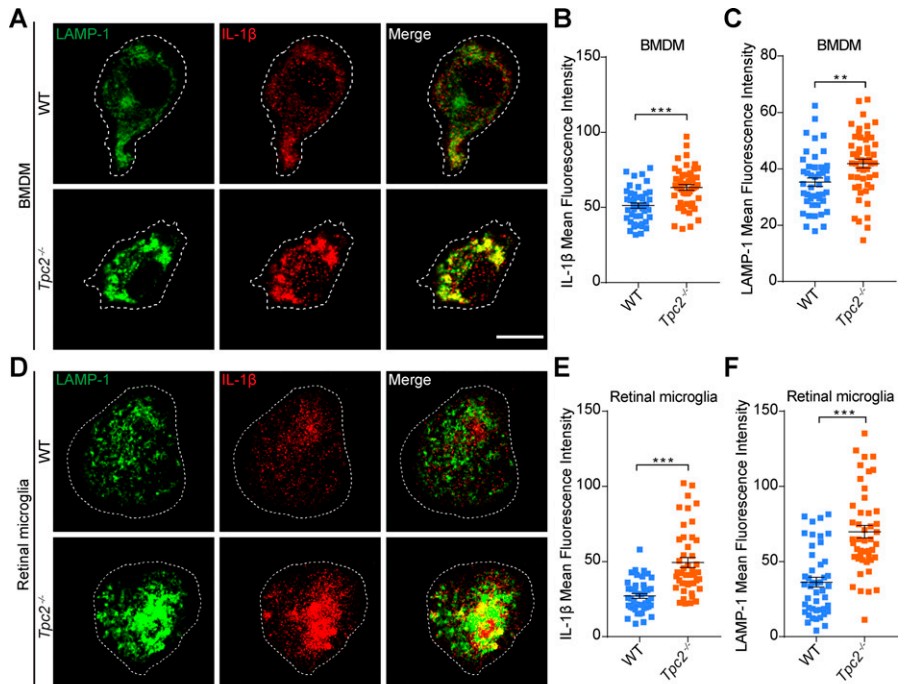

**Figure 8. Deletion of two-pore channel-2 induces intracellular accumulation of IL-1β in macrophages and retinal microglia.**
**(A, D)** Detection of IL-1β in cultured BMDMs (A) and retinal microglia (D) cells from WT and *Tpc2⁻/⁻* mice. Cells were pretreated with 1 μg/ml LPS for 4 h, followed by stimulation with 5 mM ATP for 20 min. Then cells were fixed and co-stained for the lysosomal marker protein LAMP-1 (green) and IL-1β (red). Scale bar (BMDM, A) = 10 μm, Scale bar (retinal microglia, D) = 10 μm. **(B, C, E, F)** Densitometric analysis of intracellular levels of IL-1β and LAMP-1 in BMDMs (B, C) and in retinal microglia (E, F) of WT and *Tpc2⁻/⁻* mice. Data are presented as mean ± SEM, n = 40–50 cells/group. Two-tailed *t* test, \**P* < 0.05, \*\**P* < 0.01, \*\*\**P* < 0.001.

fibroblasts. Interestingly, the global VEGFA levels in retinal tissue harvested after the laser-induced CNV were lower in *Tpc2⁻/⁻* than in WT control mice. Thus, TPC2 might also be involved in pathways that are upstream of VEGFA and also contribute to the effects observed in laser-induced CNV. Neovascular AMD is characterized by increased intraocular levels of several proangiogenic cytokines and growth factors among which VEGFA is most prominent (34). In the inflammatory process underlying neovascular AMD, VEGFA is produced by the RPE as well as by activated microglia and macrophages that are invading the area of tissue damage. Notably, deletion of TPC2 did not affect VEGFA secretion from macrophages (Fig S9). However, we consistently observed a decreased invasion of immune cells in lesion areas of *Tpc2⁻/⁻* eyes compared with WT eyes. Thus, the lower retinal VEGFA levels in *Tpc2⁻/⁻* mice after laser damage may result from the lower number of accumulated microglia and macrophages in CNV lesion areas. Moreover, using endolysosomal patch-clamp experiments we identified TPC2-mediated currents in microglia and macrophages, indicating that this ion channel is functional in these immune cells. By contrast, we only detected low levels of TPC2 mRNA in RPE and failed to detect TPC2-mediated currents in RPE cells (not shown) suggesting that TPC2 plays a secondary role in these cells.

Another major disease factor linked to AMD is the proinflammatory cytokine IL-1β. Macrophages and microglia produce and secrete high amounts of this cytokine in a complex signaling cascade that involves an initial priming step triggered by noxious signals, such as pathogen- or damage-associated molecular patterns (PAMPs or DAMPs), and activation of the inflammasome. Subsequently, the released IL-1β can trigger multiple actions, including the induction of photoreceptor degeneration (35), VEGFA release (36), CNV formation (37), and further recruitment of

macrophages and microglia. Interestingly, the lysosome has been shown to play a role in the process of inflammasome activation (38). *Tpc2⁻/⁻* mice displayed lower IL-1β levels in the laser-damaged retina suggesting that TPC2 may be involved in the generation and/or the release of this cytokine. Experiments with cultured macrophages and microglia showed that the initial priming step leading to the synthesis of inactive pro-IL-1β was not affected by the deletion of TPC2. However, the amount of mature IL-1β in cell culture supernatant was significantly lower in *Tpc2⁻/⁻* mice than in WT controls. Two additional findings could help explaining this difference. First of all, *Tpc2⁻/⁻* mice showed reduced amounts of MAC (C5b-9), which suggests reduced activation of the inflammasome. Thus, the maturation of inactive pro-IL-1β to active IL-1β could be impaired in *Tpc2⁻/⁻* mice. Second, we observed the accumulation of IL-1β in endolysosomal (LAMP-1+) organelles of *Tpc2⁻/⁻* mice, which suggests that TPC2 is involved in vesicular release of this cytokine. It has been shown that IL-1β is translated on free polyribosomes associated with the cytoskeleton in LPS-activated monocytes (39). Synthesized pro-IL-1β is mainly found in the cytosol and only a fraction resides in endolysosomal vesicles (40). When immune cells respond to a secondary stimulus such as ATP released by dying cells or by other immune cells, a large amount of pro-IL-1β is processed and subsequently secreted extracellularly (41). IL-1β is produced without a signal sequence and does not follow the conventional route of protein secretion, but rather uses nonconventional pathways of secretion, including lysosomal exocytosis (32, 42). The presence of mature IL-1β in endolysosome-enriched organelles but not in the cytosol highlights that lysosomal vesicles are the major compartment to transport processed IL-1β (43). Our data point out an essential role of TPC2 in this particular secretory pathway. However, given that a fraction of IL-1β was still secreted

extracellularly in absence of TPC2, we do not rule out the possibility that other factors or pathways also participate in IL-1β secretion. Overall, TPCs are key regulators of endolysosomal trafficking and inhibition of these channels has been shown to cause lysosomal accumulation of a variety of cargoes, including LDL/R (21), integrin (20), Ebola virus (22), and several growth factors, such as epithelial growth factor (EGF) (21) and PDGF (44). Our study adds IL-1β to the list of proteins whose release is controlled by TPC2.

Taken together, we identify TPC2 as a novel regulator of the inflammatory and neovascular processes underlying neovascular AMD. Importantly, TPC2 seems to impact different key aspects of AMD pathomechanisms (Fig 9). Our results argue that TPC2 is a key factor controlling the lysosomal release of IL-1β in macrophages and microglia, and, thereby the initiation of a number of consecutive events, including induction of cell death, macrophage recruitment, and release of VEGFA from RPE, macrophages, and other retinal cells. IL-1β has been suggested to promote photoreceptor cell death in the retina (45). In agreement with this hypothesis inhibition of IL-1β using siRNAs or neutralizing antibodies has been shown to ameliorate retinal degeneration, reducing immune cell recruitment to retinal lesion sites and production of chemokines, (12). Weaker inflammatory activity resulting from diminished IL-1β secretion could also explain the reduced complement activation (C5b-9) that was observed in $Tpc2^{-/-}$ mice.

The pathology of AMD is quite complex and comprises several other cytokines including IL-1α, IL-6, IL-18, IL-33, TNFα, and TGFβ, other factors such as complement factors and placental growth factor. It remains to be examined whether TPC2 is also involved in

controlling the secretion and signaling cascades of these proteins. Our study strongly suggests that TPC2 is a promising novel target for developing future therapies of neovascular AMD.

# Materials and Methods

### Mice

Animals were raised in 12-h light–dark cycles with food and water. $Tpc2^{-/-}$ mice were generated as described (21) and breaded on mixed Sv129/C57BL/6J genetic background. For pharmacological experiments WT C57BL/6J mice were used.

### Chemical compound preparation

Tetrandrine (Cat. no. T2695) was purchased from Sigma-Aldrich and reconstituted in DMSO as a stock concentration of 5 mM. For experiments, the tetrandrine stock solution was diluted in the choroidal sprouting culture medium with a final concentration of 5 μM or in PBS with a final concentration of 10 μM for in vivo model. Similarly, Ned-19 (Cat. no. 3954; Tocris) was dissolved in DMSO with a stock concentration of 100 mM and subsequently diluted in culture medium with a working concentration of 100 μM for sprouting culture or in PBS with a working concentration of 2 mM for in vivo study. DMSO solution with respective concentration were prepared in the same manner and served as controls.

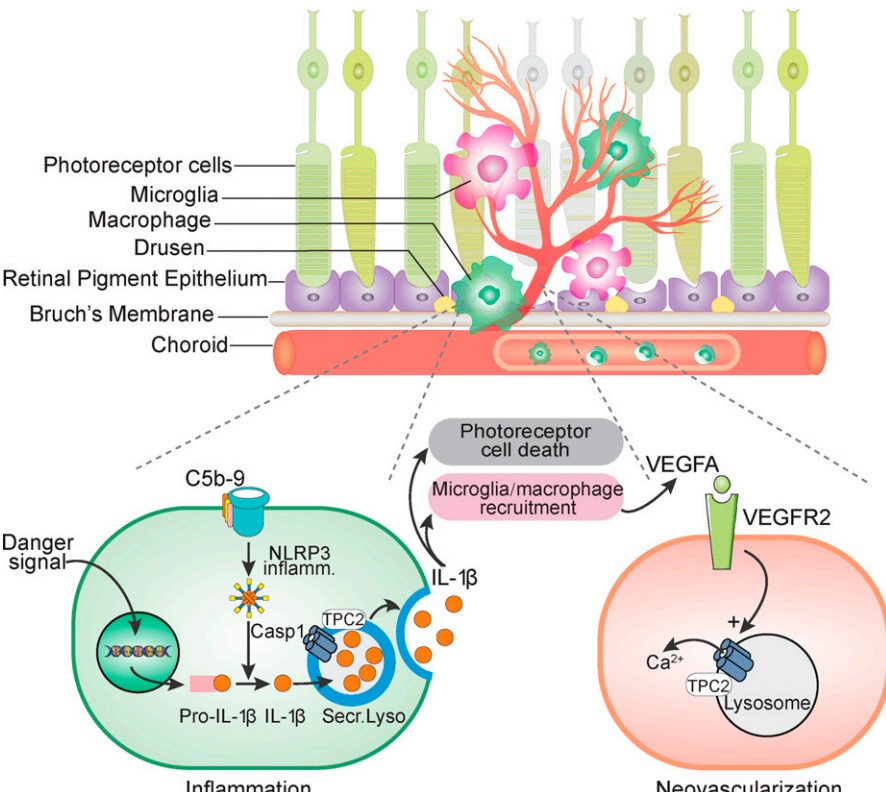

**Figure 9. Cartoon illustrates the roles of two-pore channel (TPC)-2 in the pathomechanism of CNV.** The endolysosomal cation channel TPC2 regulates exocytosis of proinflammatory cytokines such as IL-1β. Moreover, TPC2 also controls neoangiogenesis by mechanisms downstream of IL-1β. This finding is supported by previous work showing that TPC2 is involved in VEGFA signaling through VEGFR2 by providing lysosomal Ca²⁺ signals that trigger endothelial cell proliferation and vessel formation (25).

## Laser-induced choroidal neovascularization (CNV) mice model

To induce choroidal neovascularization in mice, laser coagulation was used as described ([27]). All animals used in this model were between 6 and 8 wk of age and of female gender to exclude any variance caused by gender. Briefly, mice were anesthetized with a mixture of ketamine (40 mg/kg body weight) and xylazine (20 mg/kg body weight). Pupils were dilated with tropicamide eye drops (Mydriadicum Stulln; Pharma Stulln GmbH). To keep eyes moist, hydroxypropyl methylcellulose (Methocel 2%; OmniVision GmbH) was applied on both eyes before placing mice on warm platform. Photocoagulation was performed using the image-guided laser system (Micron IV; Phoenix Research Laboratories) with the following parameters: laser wavelength, 532 nm; burn duration, 70 ms; energy, 280 mW for mixed-background $Tpc2^{-/-}$ and litter-matched WT mice and 230 mW for C57BL/6J mice. Four laser burns per eye were injected on both eyes in a distance of two optic discs from the optic disc border.

Angiographic analysis of CNV leakage was performed on day 7 and day 14 post laser coagulation. Mice were anesthetized and pupils dilated as described above. Mice received a subcutaneous injection of fluorescein sodium solution (7.5 mg/kg) to visualize vessels. Fundus fluorescein angiography (FFA) was performed 5–10 min after the injection with the retinal imaging microscope (Phoenix Research Laboratories). Leakage areas were quantified using ImageJ64 software (National Institutes of Health).

For experiments with pharmacological inhibitors, mice were anesthetized and pupils dilated as described above. Anesthetized mice were subjected to a single intravitreal injection with 1 µl of 10 µM tetrandrine or 2 mM Ned-19 into right eyes, respectively. Left eyes were used for vehicle controls. Laser injury was performed immediately after intravitreal injection and CNV leakage was assessed on day 14 after laser treatment as described above.

## Choroidal sprouting assay

Choroidal sprouting assay was carried out as previously described ([29]) with slight modifications. Briefly, pieces of the RPE/choroid/sclera complex were dissected from peripheral region of eyes of 21–25-d-old mice, cut into 1 × 1-mm fragments and then embedded in growth factor-reduced Matrigel (Cat. no. 354230; Corning) in 24-well plates. 1 ml full medium, including EBM-2 Basal Medium (Cat. no. CC-3156; Lonza), EGM-2 SingleQuots Supplements (Cat. no. CC-4176; Lonza) containing VEGFA and 1% Penicillin/Streptomycin (Cat. no. P0781; Sigma-Aldrich), was added to each well. Cultures were incubated at 37°C with 5% $CO_2$ and medium was replaced every 48 h. For WT and $Tpc2^{-/-}$ sprouting culture, RPE/choroid/sclera pieces were pre-incubated in full medium for 2 d before placing in the EBM-2 basal medium without EGM-2 SingleQuots Supplements for another 2 d. In pharmacological experiments, tissue pieces were cultured for 4 h in full medium once embedded in matrigel. Thereafter, medium was replaced with fresh medium containing 5 µM tetrandrine, 100 µM Ned-19 or vehicle controls. Images of tissue pieces were taken every day with EVOS M5000 microscope (Thermo Fisher Scientific) between day 1 and day 4. Sprouting areas were analyzed using the ImageJ64 software with SWIFT-choroid macros developed by Z Shao and M Friedlander ([29]).

## Primary culture of macrophages, RPE, and microglia

### Primary PMs

PM were obtained from 2 to 3-mo-old WT and $Tpc2^{-/-}$ mice by rinsing peritoneal cavity with cold calcium- and magnesium-free Dulbecco's phosphate-buffered saline (DPBS, Cat. no. 14190094; Gibco). After centrifugation, cell pellets were resuspended by RPMI 1640 medium (Cat. no. 21875034; Thermo Fisher Scientific) containing 10% heat-inactivated FBS (Cat. no. S0615; Sigma-Aldrich) and 1% penicillin/streptomycin and seeded in six-well culture plate or sterile 12-mm-diameter glass coverslips. For attachment, cells were incubated for 2 h at 37°C in 5% $CO_2$. Thereafter, medium was replaced with fresh medium to remove the non-adherent cells. PM were cultured for 48 h and then used for RNA extraction or endolysosomal patch-clamp experiments. The identity of PM was verified by immunostaining with the macrophage marker CD11b (Fig S5B).

### BMDMs

BMDMs were prepared as previously described ([46]). Briefly, bone marrow was flushed out from tibia and femur, which were dissected from 2 to 3-mo-old WT and $Tpc2^{-/-}$ mice, with cold PBS. After centrifugation, cell pellets were resuspended and cultured in RPMI 1640 medium supplemented with 10% heat-inactivated FBS, 40 ng/ml murine monocyte-colony stimulating factor (M-CSF, Cat. no. 130-101-703; Miltenyi Biotec) and antibiotics for 7 d at 37°C and 5% $CO_2$ to promote differentiation of cells into macrophages. The identity of BMDM was verified by immunostaining with the macrophage marker CD11b (Fig S5C). To study VEGF secretion, BMDMs were seeded in six-well plate, and then cultivated either in normal incubator or in the hypoxic chamber (Don Whitley Scientific) at 1% $O_2$. After 48 h, cell supernatant was collected and used for ELISA analysis.

### RPE

RPE cell culture was prepared as previously described ([47]). Briefly, eyeballs were dissected from 6 to 8-wk-old WT mice and placed in DMEM-F12 medium (Cat. no. 11320033; Gibco) overnight at room temperature. On the next day, eyes were applied to the enzymatic solution consisting of 2 mg/ml collagenase type 1 (Cat. no. LS004196; Worthington) and 1.5 mg/ml Trypsin (Cat. no. 27250-018; Life technologies) for 1 h at 37°C. Eyes were then placed in DMEM-F12 supplemented with 20% FBS and antibiotics to stop enzymatic reaction. RPE sheets were slightly ripped from retina and further dissociated in trypsin before seeding in 24-well plates coated with FBS. After 24 h incubation at 37°C with 5% $CO_2$, cells were carefully washed with PBS and filled with fresh medium supplemented with 10% FBS and antibiotics. Fig S5A shows a representative image of primary RPE culture.

### Microglia

For the preparation of brain microglia culture, brains were dissected from 3 to 5-d-old pups and placed into HBSS (Cat. no. 14170112; Gibco) without calcium and magnesium. Meninges were removed and brains were cut into small pieces. Tissues were dissociated in enzyme mixture of papain (Cat. no. P5306; Sigma-Aldrich) and DNase I (Cat. no. DN25; Sigma-Aldrich), subjected to

mechanical pipetting with glass pipette, and passed through 40 μM cell strainer to remove cell clumps. After centrifugation and resuspension in sorting buffer (Miltenyi Biotec), cell mixture was incubated with CD11b MicroBeads (Miltenyi Biotec) and applied directly to an LS column (Miltenyi Biotec). Microglia were isolated by Magnetic-activated cell sorting column technology according to manufacturer's instruction and resuspended with DMEM-F12 medium supplemented with 10% FBS and antibiotics. Cells were plated in 24-well plate or glass coverslips and maintained in 5% $CO_2$ at 37°C. The medium was replaced every 3–4 d.

For the culture of retinal microglia, retinas were collected from pups at early postnatal days and then delicately freed of both RPE and choroid. Dissection of eyes was performed in the presence of cold HBSS without calcium and magnesium. The tissues were later centrifuged and the cell pellet resuspended in a medium mainly containing DMEM/F-12 and heat-inactivated FBS. After gentle pipetting, the suspension was transferred to a 25 $cm^2$-flask and maintained at 37°C with 5% $CO_2$ until confluence was reached. The medium was replaced the first time after 1 wk and later every 3 d. Then cells were detached by trypsin for 5 min at 37°C to remove astrocytes and Müller cells. The remaining retinal microglia were scraped by scraper and sub-cultured in six-well plate or on 12-mm-diameter coverslips for further use.

### Interleukin-1β secretion assay

To induce IL-1β expression and secretion, BMDMs were seeded in 12-well plate to a cell density of 3 ×10$^5$/well. Culture medium was replaced with fresh medium supplemented with or without 1 μg/ml LPSs from *Escherichia coli* serotype O1101:B4 (ALX-581- 012-L001; Enzo). Cells were primed with LPS for 4 h to induce pro-IL-1β expression, followed by adding 5 mM adenosine 5′-triphosphate (ATP, A7699; Sigma-Aldrich) for 5, 10, and 20 min to stimulate the maturation and secretion of IL-1β. Extracellular medium was collected and centrifuged at 10,000*g* for 30 min to remove cell debris. The supernatant was transferred to a fresh tube for ELISA analysis. The remaining cells were washed with PBS and lysed by adding 150 μl TX buffer (0.5% Triton X-100, vol/vol) supplemented with cOmplete ULTRA Protease Inhibitor Cocktail tablets (Roche) using the mixer mill MM400 (Retsch). Cell lysate was centrifuged and supernatant was transferred to a fresh tube for Western blot analysis. For immunocytochemistry, BMDMs or retinal microglia were seeded and cultivated in 12-mm-diameter coverslips at a density of 5 ×10$^4$/well or 2.5 × 10$^4$/well. Cultures were treated with 1 μg/ml LPS for 4 h, followed by stimulation with 5 mM ATP for 20 min as above. After washing with PBS, cells were fixed with 4% PFA and used for immunocytochemistry experiments.

### Endolysosomal patch-clamp experiments

Protocols for whole-endolysosomal recording have been described previously in detail (48). In brief, the cells were treated with vacuolin (Cat. no. sc-216045; Santa Cruz Biotechnology) at 37°C and 5% $CO_2$ for 1 h. Vacuolin was washed out at least 10 min before patch-clamp experimentation. Isolation-micropipettes were used to open up the plasma membrane of macrophages on the coverslips, and isolation of the vesicle out of the cell. Afterwards,

electrode-pipettes were applied to form whole-endolysosomal configuration for patch-clamping.

Macrophages and microglia were used for experiments within 2–10 d after preparation. Patch recordings were performed with an Axopatch 200B amplifier (Molecular Device) and a Digidata 1320A data acquisition system (Molecular Device). PClamp and Clampfit software were used to record and analyze data. Recording glass pipettes were polished and had a resistance of 4–8 MΩ. For all experiments, salt-agar bridges were used to connect the reference Ag-AgCl wire to the bath solution to minimize voltage offsets. Liquid junction potential was corrected. Cytoplasmic solution contained 140 mM K-MSA, 5 mM KOH, 4 mM NaCl, 0.39 mM $CaCl_2$, 1 mM EGTA, and 10 mM Hepes (pH was adjusted with KOH to 7.2). Luminal solution contained 140 mM Na-MSA, 5 mM K-MSA, 2 mM Ca-MSA, 1 mM $CaCl_2$, 10 mM Hepes, and 10 mM MES (pH was adjusted with NaOH to 4.6). 1 μM PI(3,5)$P_2$ (water-soluble diC8 form, from Echelon Biosciences) was applied to evoke endogenous TPC2 currents on isolated endolysosomes (19). 500 ms voltage ramps from –100 to +100 mV were applied every 5 s, holding potential at 0 mV. The current amplitudes at –100 mV were extracted from individual ramp current recordings. All statistical analyses were performed using Origin 8 software.

### Immunohistochemistry

Mice were euthanized and their eyes were enucleated and fixed in 4% PFA. Subsequently, cornea, retina, and lens were removed from the eyes. The remaining RPE–choroid complex was flat-mounted and stained with Alexa Fluor 488–conjugated *Griffonia (Bandeiraea) simplicifolia* IB4 (1:25, I21411; Thermo Fisher Scientific), rabbit anti-Iba-1 (1:500, 019-19741; Wako), or rabbit anti-C5b-9 (1:500, ab55811; Abcam) overnight at 4°C and then incubated with secondary antibodies Alexa Fluor 555 donkey anti-rabbit immunoglobulin G (IgG) (1:500, A-31572; Invitrogen) or Alexa Flour 488 donkey anti-rabbit IgG (1:500, A21206; Invitrogen) for 2 h at room temperature. Cell nuclei were visualized with Hoechst 33342 solution (5 μg/ml; Invitrogen, Thermo Fisher Scientific). To detect IL-1β expression in the retina, mice eyes from WT and *Tpc2$^{-/-}$* mice were enucleated and processed after 24 h of laser treatment. Eyes were fixed in 4% PFA and dehydrated with 30% sucrose. They were embedded in optimum cutting temperature compound and cryosectioned at a thickness of 10 μm. Sections were stained with guinea pig anti–Iba-1 antibody (1:500, 234004; Synaptic System) and mouse anti–IL-1β (1:50, 12242S, or 1:500; Cell Signaling Technology, ab9722; Abcam) overnight, followed by incubation with secondary antibodies Flourescein (FITC) donkey anti-guinea pig IgG (1:400, 706-095-148; Jackson) and Alexa Fluor 647 goat anti-mouse IgG (1:500, A21236; Life Technologies) for 2 h. Cell nuclei were visualized with Hoechst 33342 solution. Images were obtained using the Leica TCS SP8 spectral confocal laser scanning microscope (Leica), acquired with LASX software, and quantified with ImageJ64 software.

### Immunocytochemistry

PMs, BMDMs, or retinal microglia were cultured onto sterile 12-mm-diameter coverslips. Cells were washed with PBS followed by fixation in 4% PFA for 10 min and blocking in 5% ChemiBlocker (Merck

**Life Science Alliance**

Millipore) and 0.1% Saponin (Cat. no. 47036; Sigma-Aldrich) for 1 h. Thereafter, cells were incubated with rat anti-CD11b (1:500, 101202; BioLegend), rabbit anti–IL-1$\beta$ (1:500, ab205924; Abcam), or rat anti-LAMP-1 (1:120, sc-19992; Santa Cruz Biotechnology) overnight at 4°C, followed by a 2-h incubation with goat anti-rabbit IgG conjugated with Alexa Fluor 647 (1:500, A32733; Life Technologies) or donkey anti-rat IgG conjugated with Alexa Fluor 488 (1:500, A21208; Thermo Fisher Scientific). Cell nuclei were visualized with Hoechst 33342 solution. Images were obtained using the Leica TCS SP8 spectral confocal laser scanning microscope (Leica), acquired with LASX software, and quantified by ImageJ64 software. All images within each experimental series were collected and processed with the same microscope settings and parameters.

For staining of choroidal sprouts, choroidal explants were washed with PBS. 4% PFA was then added for 60 min before treatment with rabbit anti–NG2 chondroitin sulfate proteoglycan antibody (1:250, AB5320; Sigma-Aldrich), and rabbit anti-vimentin (1:100, 5741; Cell Signaling Technology) overnight at 4°C. Thereafter, the secondary antibody Alexa Fluor 555 donkey anti-rabbit IgG was applied overnight at 4°C. Cell nuclei were visualized with Hoechst 33342 solution. Images were collected from z-stack using the Leica TCS SP8 spectral confocal laser scanning microscope and processed with maximum projection.

### ELISA

48 h after laser treatment, eyes were enucleated from WT and $Tpc2^{-/-}$ mice. Retinas harvested from the eyes were placed and homogenized in cold PBS containing protease inhibitor. After centrifugation, the upper supernatant was collected for ELISA. The level of VEGFA protein in the supernatant was determined using a MILLIPLEX MAP Mouse Cytokine/Chemokine Magnetic Bead Panel–Premixed 32 Plex–Immunology Multiplex Assay (Cat. no. MCYTMAG-70K; MERCK). VEGFA in cell culture medium of BMDMs were detected with Mouse VEGF Quantikine ELISA Kit (MMV00). ELISA was performed according to the manufacture's protocol. The concentration of VEGFA was calculated using standard curves obtained from solutions with known VEGFA concentration and then normalized to total protein. To determine IL-1$\beta$ protein level in the eyes, retinas were carefully separated from the eyes after 24 h of laser injury, homogenized in cold PBS containing protease inhibitor. After centrifugation at 4°C, the supernatant was collected and analyzed for total protein content by Qubit Protein Assay kits (Q33211; Life Technologies) and for IL-1$\beta$ level with a mouse IL-1$\beta$/IL-1F2 Quantikine ELISA kit (MLB00C; R&D system). ELISA was performed according to the manufacture's instruction. The concentration of IL-1$\beta$ was calculated using standard curves and normalized to the total protein level. To detect IL-1$\beta$ concentration in extracellular medium, the total volume of 800 μl fresh medium was added into each well, then BMDMs were challenged with or without 1 μg/ml LPS for 4 h and 5 mM ATP for various time. Cell culture supernatant was centrifuged to remove cell debris and proceeded for the measurement of IL-1$\beta$ concentration with ELISA as described above. The concentration of IL-1$\beta$ was calculated using standard curves.

### Western blot analysis of IL-1$\beta$

BMDM cell lysate containing 15 μg protein was boiled in 1× Laemmli sample buffer supplemented with DTT at 72°C for 10 min. The proteins were fractionated by SDS–PAGE on a 6–12% gradient gel. The separated proteins were transferred to a PVDF membrane (Millipore) and subjected to immunoblotting with antibodies against IL-1$\beta$ (1:1,000, ab9722; Abcam) and $\beta$-tubulin (1:1,000, 86298; Cell Signaling Technology). Densitometric analysis of blots was performed using the ImageLab software (Bio-Rad).

### Quantitative RT–PCR analysis

RNA from cultured cell was extracted using RNeasy Plus Mini kit (QIAGEN) according to the manufacturer's instructions. cDNA was synthesized with the RevertAid First Strand cDNA Synthesis Kit (Thermo Fisher Scientific). qRT-PCR was performed by using SYBR Select Master Mix (Applied Biosystems) on a StepOnePlus Real-time PCR System (Applied Biosystems). The following primers were used: TPC2, forward (fw): 5′-TGCTGCAGAATTCCTCCATGAT-3′; reverse (rv): 5′-ATGGTGAAGAGACACAGGTGG-3′. IL-1$\beta$, fw: 5′-TGCCACCTTTTGACAGTGATGA-3′; rv: 5′-ATCAGGACAGCCCAGGTCAA-3′. Hypoxanthine guanine phosphoribosyl transferase (Hprt), fw: 5′-TGCCGAGGATTTGGAAAAAGTG-3′; rv: 5′-TGGCCTCCCATCTCCTTCAT-3′.

### Statistics

All graphical data are represented as mean ± SEM for the indicated n values of the experiments. Statistical analysis was carried out using GraphPad Prism 7 or Origin 8 software. $P$-values were calculated by two-tailed $t$ test, two-way ANOVA, or one-way ANOVA with Tukey's multiple comparisons test. $P < 0.05$ were considered to show significant differences, *$P < 0.05$, **$P < 0.01$, ***$P < 0.001$, ns, no significant differences.

### Study approval

Animal experiments were approved by the District Government of Upper Bavaria in accordance with the German Animal Welfare and Institutional guidelines.

# Data and Materials Availability

All data needed to evaluate the conclusions in the paper are present in the paper and/or the Supplementary Materials.

# Supplementary Information

# Acknowledgements

We thank Maximilian Gerhardt (Eye Hospital, LMU Munich), Olaf Strauß (Charité Berlin), and Elisabeth Butz (Center for Genomic Medicine, Department of Neurology, Massachusetts General Hospital, Harvard Medical School, Boston, MA, USA) for scientific and/or experimental advice. Y Li was supported by a scholarship of the Chinese Scholarship Council. M Biel,

S Michalakis, C Grimm, and C Wahl-Schott were supported by the Deutsche Forschungsgemeinschaft (DFG, German Research Foundation) TRR-152/2 Projects 12 (to M Biel) 4 (to C Grimm) and 6 (to C Wahl-Schott), KL 1119/6-1 (to N Kligbauer) and EXC114 (to M Biel and S Michalakis).

## Author Contributions

Y Li: data curation, validation, investigation, visualization, methodology, and writing—original draft, review, and editing.
C Schön: data curation, investigation, and methodology.
C-C Chen: data curation, investigation, and methodology.
Z Yang: methodology.
R Liegl: supervision, investigation, and methodology.
E Murenu: data curation, investigation, and methodology.
B Schworm: data curation.
N Klugbauer: methodology.
C Grimm: methodology.
C Wahl-Schott: funding acquisition and methodology.
S Michalakis: supervision, funding acquisition, validation, project administration, and writing—original draft, review, and editing.
M Biel: supervision, funding acquisition, validation, project administration, and writing—original draft, review, and editing.

## Conflict of Interest Statement

The authors declare that they have no conflict of interest.

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
