## [Reviewer comments · Life Science Alliance]

Life Science Alliance

TPC2 promotes choroidal angiogenesis and inflammation in a mouse model of neovascular AMD.

Yanfen Li, Christian Schön, Cheng-Chang Chen, Zhuo Yang, Raffael Liegl, Elisa Murenu, Benedikt Schworm, Norbert Klugbauer, Christian Grimm, Christian Wahl-Schott, Stylianos Michalakis, and Martin Biel

DOI: <https://doi.org/10.26508/lsa.202101047>

Corresponding author(s): Stylianos Michalakis, University Hospital, LMU Munich and Martin Biel, Pharmakologie für Naturwissenschaften, Ludwig-Maximilians-Universität München

Review Timeline:

Submission Date:	2021-02-05
Editorial Decision:	2021-03-26
Revision Received:	2021-05-07
Editorial Decision:	2021-06-08
Revision Received:	2021-06-14
Accepted:	2021-06-15

Transaction Report:

March 26, 2021

Re: Life Science Alliance manuscript #LSA-2021-01047-T

Prof. Stylianos Michalakis
Ludwig-Maximilians-Universität München
Ophthalmology
Mathildenstr. 8
Munich, Bavaria 80336
GERMANY

Dear Dr. Michalakis,

Thank you for submitting your manuscript entitled "TPC2 promotes choroidal angiogenesis and inflammation in a mouse model of neovascular age-related macular degeneration" to Life Science Alliance. The manuscript was assessed by expert reviewers, whose comments are appended to this letter. We would like to invite you to submit a revised version of the manuscript to Life Science Alliance.

We apologize for this unusual and extended delay in getting back to you. As you will note from the reviewers' comments below, the reviewers are excited about these findings, but have also raised some reasonable questions and concerns that should all be addressed in the revised manuscript.

Thank you for this interesting contribution to Life Science Alliance. We are looking forward to receiving your revised manuscript.

Sincerely,

Shachi Bhatt, Ph.D.

Executive Editor

Life Science Alliance

<https://www.lsjournal.org/>

Interested in an editorial career? EMBO Solutions is hiring a Scientific Editor to join the international Life Science Alliance team. Find out more here -

https://www.embo.org/documents/jobs/Vacancy_Notice_Scientific_editor_LSA.pdf

B. MANUSCRIPT ORGANIZATION AND FORMATTING:

Reviewer #1 (Comments to the Authors (Required)):

This is an important paper demonstrating a major role of the endolysosomal channel TPC2 in the mechanisms underlying neovascular age-related macular dystrophy (AMD). Using a mouse model (laser-induced choroidal neovascularization) of neovascular AMD, the authors

use *Tpcn2*^{-/-} mice to demonstrate a protective effect in cellular and biochemical changes in this model of the disease. This is phenocopied by the use of selective pharmacological inhibitors of TPC2 (tetrandrine/Ned19). Importantly, the effect was specific for TPC2 over TPC1.

A strength of the study is the use of ex-vivo retinal preparations in their studies.

Overall, the data supports their conclusions and the statistical analyses are appropriate. The paper is well-written and clear.

I just have minor points:

1. The concentration of Ned19 used is high (2 mM), but this may be needed in the tissue preparations. Some comment or concentration-effect data would be helpful.
2. The authors have developed two agonists of TPC2, TPCA1N and TPCA1P. These respectively activate the two different modes of channel function, calcium release and sodium fluxes, respectively. Which mode do they think is the most important for the effects here eg IL1 β exocytosis?
3. The reduced levels of VEGFA are interesting in the KO mice, although it was previously shown that VEGFR2 signalling was affected. This was briefly mention in the discussion, but can the authors expand on this a little?
4. Do *Trpm1*^{-/-} mice (another lysosomal cation channel) show similar effects to the *Tpcn2*^{-/-} mice?
5. The lysosomal ion currents in brain microglial cells are barely above basal in contrast to macrophages (Fig. 5). Can the authors comments on this? In the macrophages, can they show tetrandrine/Ned19 block the currents (and what concentrations are required?)
6. A number of references could be added:
Davis et al (2020) *Embo J* 39:e104058 have shown TPCs are required for macrophage phagocytosis. This could be cited on p.3.

There is a literature of lysosomal calcium release and inflammasome activation that could be cited eg Weber and Schilling 2014 *JBC* 289, 9158-9171.

Reviewer #2 (Comments to the Authors (Required)):

In this article, Li et al. investigate the role of TPC2 in the neovascular AMD, their data is clearly showing the involvement of TPC2 in their models.

Major comments:

1. The result of choroidal sprouting is got in vitro by adding VEGFA in the medium, so it is clearly independent with the following data of the decreased level of VEGFA in retinas.

For the first part of choroidal sprouting, although the authors discussed a reference of TPC2 involvement in VEGFR2 signaling, it would be better the authors could provide some evidence to link it with their result. And I'm wondering what's the main cell type of the sprouting choroid? Are those mainly RPE? If so, it is inconsistent with the result of no TPC2 in RPE. Can the authors make this result clearer?

For the 2nd part of reduced VEGFA, it is feasible the authors could provide some data of the release of VEGFA like what they did for the IL1 β release?

2. The authors proposed a model that lack of TPC2 trapped the IL1 β in macrophages and microglia. If so, they should observe a stronger co-localization of IL1 β and Iba1 in their Fig6. Can the authors quantify this and provide some discussion?

Minor Comments:

In Fig1, I'm wondering if TPC2 KO will affect the fully recover of the laser photocoagulation model. That means if the authors wait longer time, will they see a difference?

Also for Fig1, the authors don't have to show the data, but just curious to know if they have also tested the TPC1/2 double KO mice? (not necessary)

In Fig6, what's the blue signal? I guess it's DAPI? If so, why it seems some Iba1 positive cells are not blue positive?

Sentence '...and stained endothelial cells (ECs) and macrophage/microglia with the marker isolectin B4 (IB4) and the macrophage/microglia marker Iba-1 (Fig. 4A,.....', there is redundant 'macrophage/microglia'.

Reviewer #3 (Comments to the Authors (Required)):

In this manuscript, the authors examined the role of TPC2 in the release of IL1 β , recruitment of inflammatory cells, and the effect on the laser-induced CNV using TPC2 knockout mice and inhibitors for TPC2. Overall, it is a novel and interesting manuscript. While of broad interest, this manuscript needs additional revision to better explain the experimental design in order to allow the results to be better interpreted. My specific questions are raised below.

Major comments.

Query 1) In the laser-induced CNV model used in Figures 1 and 2, the authors used vascular leakage of fluorescein, however, the standard method of quantification in the mouse model is the area of IB4 staining on the choroidal flat-mount (as the authors showed in Fig. 4 for Tpc2 -/-). Please explain why the authors used fluorescein instead of IB4 staining to quantify the data. Also, the data should be presented as the average per mouse in the graph.

Query 2) In the choroidal sprouting assays in Figure 3, the authors showed N=8 pieces/group in B and 9-12 pieces/group in C, D. How many mice were used per group? As choroidal tissue is dissected from healthy eyes for this assay, there are only a few microglia or macrophages in the tissue (see Fig. S6). Although the authors showed the reduced amount of VEGF in Tpc2-/- after laser burn (Fig. 3E), it is likely due to the difference in the number of microglia/macrophages in WT and Tpc2 -/-. The explanation of the defect in the choroidal sprouting was not clear.

Query 3) Accumulation of macrophage started early time point, 1~3 days after laser burn. As the authors measured VEGF (secreted mainly from macrophage) at 2 days after laser-burn in Fig. 3E, the authors are well aware of that. However, the authors stained Iba-1 and C5b-9 at 14 days after laser burn. Please explain the rationale. Is there no difference in the early time point as the authors showed in Fig. 6B? Please quantify the Iba-1 positive cells in the early time point too. How early can the difference in the number of macrophage accumulation be seen? Is it before the authors observed the difference in the amount of VEGF (at 2 days after laser burn)?

Query 4) The interpretation of the data in Fig. 6B is not clear. The authors described "the overall intensity of the IL1- β signal is lower in Tpc2-/- compared to WT. Iba-1+ cells infiltrating the ONL and OS seem to contain higher IL-1 β amounts in Tpc2-/- than in WT mice." But the representative images do not support that. Please quantify the intensity of IL1 β in overall and in Iba-1+ cells.

LSA-2021-01047-T_R1 Point-to-point response letter

We thank all three reviewers for their comments, which helped to substantially improve the quality of the manuscript. We have carefully addressed each point raised as outlined below. Changes made in the manuscript and the supplementary information are always marked in red.

Reviewer #1 (Comments to the Authors (Required)):

This is an important paper demonstrating a major role of the endolysosomal channel TPC2 in the mechanisms underlying neovascular age-related macular dystrophy (AMD).

*Using a mouse model (laser-induced choroidal neovascularization) of neovascular AMD, the authors use *Tpcn2*^{-/-} mice to demonstrate a protective effect in cellular and biochemical changes in this model of the disease. This is phenocopied by the use of selective pharmacological inhibitors of TPC2 (tetrandrine/Ned19). Importantly, the effect was specific for TPC2 over TPC1.*

A strength of the study is the use of ex-vivo retinal preparations in their studies. Overall, the data supports their conclusions and the statistical analyses are appropriate. The paper is well-written and clear.

I just have minor points:

- 1. The concentration of Ned19 used is high (2 mM), but this may be needed in the tissue preparations. Some comment or concentration-effect data would be helpful.*

*In the experiments of Fig. 2 we injected 1 µl of a 2 mM solution of Ned-19 intravitreally. If one assumes a total vitreous volume of 4 – 5 µl the final concentration of Ned 19 would be 400-500 µM which is still quite high. In pilot experiments with lower concentrations we did not see a robust Ned-19 *in vivo* effect on neovascularization. By contrast, in the *in vitro* sprouting assay (Fig. 3D) we consistently observed a clear inhibitory effect of Ned-19 at a concentration of 100 µM. At this point, we do not know the reason for this discrepancy but bioavailability/pharmacokinetic factors impacting absorption and/or tissue penetration may underlie the lower *in vivo* efficacy of Ned-19 as compared to tetrandrine.*

- 2. The authors have developed two agonists of TPC2, TPCA1N and TPCA1P. These respectively activate the two different modes of channel function, calcium release and sodium fluxes, respectively. Which mode do they think is the most important for the effects here eg *IL1beta* exocytosis?*

This is indeed a very interesting question. Previous studies demonstrated that an increase in cytosolic Ca²⁺ is essential for ATP-induced IL-1β secretion in

macrophages [1, 2]. Moreover, it was found that lysosomal Ca^{2+} signaling, but not ER-derived and mitochondrial Ca^{2+} , regulate IL-1 β secretion by mediating lysosomal exocytosis in monocytic cells. In particular, Ca^{2+} mobilization from the lysosome could be blocked by the TPC blocker Ned-19 [3]. Collectively, it is quite likely that a TPC2-mediated Ca^{2+} signal is involved in secretion of IL-1 β in macrophages and microglia. Thus, we agree with this reviewer and believe that future follow-up studies, e.g. with specific TPC agonists that activate different permeation modes of these channels, are justified in order to test this hypothesis.

3. The reduced levels of VEGFA are interesting in the KO mice, although it was previously shown that VEGFR2 signalling was affected. This was briefly mention in the discussion, but can the authors expand on this a little?

We appreciate this comment and are happy to provide the following further clarification. We observed that a smaller number of Iba-1 $^{+}$ immune cells invaded the lesion area of *Tpc2* $^{-/-}$ eyes compared to WT eyes (Fig. 4A). It is well established that these immune cells secrete VEGFA upon activation [4]. Thus, reduced levels of VEGFA may result from the lower immune activation in *Tpc2* $^{-/-}$ retina. We have added this information in the results section on page 10 of the revised manuscript.

4. Do *Trpml1* $^{-/-}$ mice (another lysosomal cation channel) show similar effects to the *Tpcn2* $^{-/-}$ mice?

We thank Reviewer 1 for this important suggestion. We indeed performed some preliminary laser coagulation experiments in *TRPML1* $^{-/-}$ mice. However, no significant difference was found between WT and *TRPML1* $^{-/-}$ mice (see below). Since the focus of the present study is on TPCs and further clarification of any contribution of TRPML1 would require additional experiments we prefer not to include TRPML1 data in the present manuscript.

Genetic deletion of TRPML1 has no effect on choroidal neovascularization (CNV) in laser-induced mice model. Representative fundus images on day 0 and fluorescein angiography (FFA) of subretinal lesion from 6 to 8-week-old *TRPML1* $^{-/-}$ mice (A, lower panel) and WT mice (A, upper panel) 7 days (D7) after laser damage.

(B) Quantification of vascular leakage areas by analyzing pixel intensities after laser-induced damage for WT versus *TRPML1*^{-/-} mice on D7. Each data point represents one lesion site. Data are presented as mean ± SEM. Group size as n = 3 to 4 mice. 2-tailed Student's *t*-test was used for statistical analysis. ns, no significant.

5. The lysosomal ion currents in brain microglial cells are barely above basal in contrast to macrophages (Fig. 5). Can the authors comments on this? In the macrophages, can they show tetrandrine/Ned19 block the currents (and what concentrations are required?)

While a number of lysosomal ion channels (including TMEM175, BK, and TPCs) could principally contribute to the basal current in microglia, the molecular correlate of this current is still not fully understood. Tetrandrine and Ned-19 are very valuable tools for studying TPC function, however they only inhibit 50-60% of the total PI(3,5)P₂-evoked currents. Thus, these compounds are not optimal for quantifying reductions of currents in cells with very low current density such as microglia. Nevertheless, we have used 1 μM Ned-19 to challenge BMDMs and – as shown in the figure below – there is indeed a small but clearly detectable reduction of the PI(3,5)P₂ evoked current.

Isolation of TPC-currents in BMDMs using Ned-19. Representative current-voltage relationships recorded from vacuolin-enlarged endolysosomal vesicles isolated from cultured WT BMDM. Currents were recorded before and after application of 1 μM phosphatidylinositol 3,5-bisphosphate (PI(3,5)P₂) and after addition of 1 μM Ned-19.

6. A number of references could be added:
Davis et al (2020) Embo J 39:e104058 have shown TPCs are required for macrophage phagocytosis. This could be cited on p.3.

There is a literature of lysosomal calcium release and inflammasome activation that could be cited eg Weber and Schilling 2014 JBC 289, 9158-9171.

We thank the reviewer for point this out and added the missing references in the revised manuscript on Page 3 and Page 11.

Reviewer #2 (Comments to the Authors (Required)):

In this article, Li et al. investigate the role of TPC2 in the neovascular AMD, their data is clearly showing the involvement of TPC2 in their models.

Major comments:

1. The result of choroidal sprouting is got in vitro by adding VEGFA in the medium, so it is clearly independent with the following data of the decreased level of VEGFA in retinas.

For the first part of choroidal sprouting, although the authors discussed a reference of TPC2 involvement in VEGFR2 signaling, it would be better the authors could provide some evidence to link it with their result.

We agree with this reviewer that it would indeed be interesting to study the signaling cascade downstream of VEGFR2 in choroidal ECs. On the other hand, downstream effects had been previously examined by Favia et al. [5] in HUVEC, a frequently used cell model in angiogenesis research. In the present study, we focused on the elucidation of the role of TPC2 in the CNV pathology a common feature of wet AMD – which also involves activation and infiltration of immune cells – and explored so far unknown TPC2-linked effects that are upstream of VEGFR2, in particular effects of TPC2 on IL-1 β secretion from macrophages/microglia and their impact on CNV and retinal inflammation.

And I'm wondering what's the main cell type of the sprouting choroid? Are those mainly RPE? If so, it is inconsistent with the result of no TPC2 in RPE. Can the authors make this result clearer?

The sprouting culture consist of two parts, the embedded eye cup fragments which contain RPE, choroid and sclera; and microvascular sprouts. It is known from the literature that vascular sprouts are formed by ECs and surrounding pericytes and fibroblasts [6, 7]. To further address this issue we have performed immunostainings of choroidal sprouting cultures with different cell markers. The data indeed show that the sprouts are formed by ECs and also contain pericytes and fibroblasts. We did not observe RPE-like cells in the sprouts. We have included the new data in the supplement (Fig. S4) and have amended the text on Page 5.

For the 2nd part of reduced VEGFA, it is feasible the authors could provide some data of the release of VEGFA like what they did for the IL1b release?

It has been reported that hypoxia is a key trigger of VEGFA release from cultured BMDM [8]. Based on this finding, we investigated VEGFA secretion from WT and *Tpc2*^{-/-} BMDMs under normoxia or hypoxia condition. The VEGFA levels were comparable in supernatants from WT and *Tpc2*^{-/-} BMDMs as shown in the quantification plot below. Thus, while TPC2 is clearly involved in IL-1 β release, it does not seem to play a major role in secretion of VEGFA. This finding might be best explained by the fact VEGF contains a secretory signal sequence and is secreted through the endoplasmic reticulum (ER)-Golgi pathway which is independent from TPC2.

TPC2 has no effect on VEGFA secretion. WT and *Tpc2*^{-/-} BMDMs were seeded in 6-well plate and then were incubated either in normal incubator or in the hypoxic chamber (1% O₂) [9]. After 48 h, cell supernatant was collected and analyzed with ELISA. Data are presented as mean \pm SEM. Two-way ANOVA ns, no significant difference.

2. The authors proposed a model that lack of TCP2 trapped the IL1b in macrophages and microglia. If so, they should observe a stronger co-localization of IL1b and Iba1 in their Fig6. Can the authors quantify this and provide some discussion?

We thank this reviewer for the valuable suggestion. We now quantified the IL-1b mean intensity in Iba1 positive cells. We found that the mean intensity of IL-b in Iba-1 positive cells is increased in *Tpc2*^{-/-} compared to WT retina, which further corroborates our other findings. These novel data our now included in the revised Fig. 6B-C.

Minor Comments:

In Fig1, I'm wondering if TPC2 KO will affect the fully recover of the laser photocoagulation model. That means if the authors wait longer time, will they see a difference?

Laser-induced CNV is a self-limiting model of angiogenesis/inflammation and the lesions induced by the laser damage develop from day 1 to day 7, peak at days 7-10 and regress after 14 days. Following the reviewer's suggestion, we quantified the leakage area at day 14. Confirming our results obtained at day 7, also at his time-point *Tpc2*^{-/-} mice show a reduced leakage area. Given the self-limiting nature of the CNV model, we decided not to extend the in life analysis beyond day 14, but rather opted to terminate the mice and analyze the treatment effects in the *ex vivo* RPE/choroid flat mounts (data shown in Fig. 4).

Also for Fig1, the authors don't have to show the data, but just curious to know if they have also tested the TPC1/2 double KO mice? (not necessary)

TPC1/2 double knockout mice are difficult to breed, and if at all, produce very small litters. Therefore, we did not test these mice in the CNV model.

In Fig6, what's the blue signal? I guess it's DAPI? If so, why it seems some Iba1 positive cells are not blue positive?

The blue signal was Hoechst 33342 nuclear staining. The images were acquired and scanned on a single plane with confocal microscopy. Therefore, the nuclei may not always be in the same plane as other portions of the section and, thus, some Iba1 positive cells are not blue positive.

Sentence '...and stained endothelial cells (ECs) and macrophage/microglia with the marker isolectin B4 (IB4) and the macrophage/microglia marker Iba-1 (Fig. 4A,.....', there is redundant 'macrophage/microglia'.

We thank the reviewer for pointing this out and amended the text accordingly.

Reviewer #3 (Comments to the Authors (Required)):

In this manuscript, the authors examined the role of TPC2 in the release of IL1 β , recruitment of inflammatory cells, and the effect on the laser-induced CNV using TPC2 knockout mice and inhibitors for TPC2. Overall, it is a novel and interesting manuscript. While of broad interest, this manuscript needs additional revision to better explain the experimental design in order to allow the results to be better interpreted. My specific questions are raised below.

Major comments.

Query 1) In the laser-induced CNV model used in Figures 1 and 2, the authors used vascular leakage of fluorescein, however, the standard method of quantification in the mouse model is the area of IB4 staining on the choroidal flat-mount (as the

authors showed in Fig. 4 for *Tpc2*^{-/-}). Please explain why the authors used fluorescein instead of IB4 staining to quantify the data.

We agree with reviewer 3 that IB4 staining of choroidal flat mount preparations is a frequently used method for CNV quantification. However, FFA has been also extensively used in addition to flat mount stainings for this purpose [10]. Importantly, a comparative study showed that quantifications obtained with FFA correlate well with conventional area measurements from IB4 stained flat mounts [11]. Unlike postmortem analyses, FFA quantifications allow for determination of time courses of CNV in a given individual and are hence well suited for explorative studies. To demonstrate the principal role of TPC2 in CNV we have therefore chosen FFA-based quantification as initial method in Figs. 1 and 2. However, we also used RPE/choroid flat mount stainings (see Fig. 4), which confirmed the effects on neovascularization seen with *in vivo* FFA and also allowed us to analyze infiltration of immune cells in the leakage area. In agreement with the FFA data, *TPC2*^{-/-} mice showed significantly weaker IB4 staining than WT controls confirming that FFA and IB4 stainings yield comparable results.

Also, the data should be presented as the average per mouse in the graph.

We agree with the reviewer that the data can be presented as the average per mouse. However, there are also studies based on cumulative analysis of lesion areas (e.g. [12]). Since both methods are equivalent if the mouse number is sufficiently high (as in our study), we would like to stay with our quantification method.

Query 2) In the choroidal sprouting assays in Figure 3, the authors showed N=8 pieces/group in B and 9-12 pieces/group in C, D. How many mice were used per group? As choroidal tissue is dissected from healthy eyes for this assay, there are only a few microglia or macrophages in the tissue (see Fig. S6). Although the authors showed the reduced amount of VEGF in *Tpc2*^{-/-} after laser burn (Fig. 3E), it is likely due to the difference in the number of microglia/macrophages in WT and *Tpc2*^{-/-}. The explanation of the defect in the choroidal sprouting was not clear.

We used 1-2 mice for each group in choroidal sprouting culture.

Reviewer #2 had a similar comment regarding the mechanism underlying reduced sprouting in *Tpc2*^{-/-} mice (please see above for our detailed response to question #1 of reviewer #2). Briefly, we believe that the reduced sprouting in choroidal tissue culture in the presence of a fixed concentration of VEGFA is reflecting the fact that TPC2 is involved in downstream signaling of VEGFR2 as published previously by Favia et al. [5]. In contrast to IL-1 β , VEGFA secretion from macrophages was not different between WT and *Tpc2*-deficient mice. Thus, and as assumed by the reviewer, the reduction of VEGFA in the retina of *Tpc2*^{-/-} mice is probably due to the difference in the number of microglia/macrophages in WT and *Tpc2*^{-/-} rather than a defect in secretion.

Query 3) Accumulation of macrophage started early time point, 1~3 days after laser burn. As the authors measured VEGF (secreted mainly from macrophage) at 2 days after laser-burn in Fig. 3E, the authors are well aware of that. However, the authors stained Iba-1 and C5b-9 at 14 days after laser burn. Please explain the rationale. Is there no difference in the early time point as the authors showed in Fig. 6B? Please quantify the Iba-1 positive cells in the early time point too. How early can the difference in the number of macrophage accumulation be seen? Is it before the authors observed the difference in the amount of VEGF (at 2 days after laser burn)?

We can confirm that invasion of macrophages to lesion areas starts at a very early time point. To further explore this, we counted the number of Iba-1 positive cells in ONL and OS one day after laser injury. The corresponding quantification plot has been added to the revised manuscript as novel Fig. S7B. The data show less pronounced accumulation of Iba-1 positive cells in the ONL and OS areas of *Tpc2*^{-/-} retina compared to WT. This finding is consistent with a model explaining the reduced VEGFA levels of *Tpc2*-deficient mice by a lower number of activated macrophages in the CNV lesion areas.

Query 4) The interpretation of the data in Fig. 6B is not clear. The authors described "the overall intensity of the IL1- β signal is lower in *Tpc2*^{-/-} compared to WT. Iba-1+ cells infiltrating the ONL and OS seem to contain higher IL-1 β amounts in *Tpc2*^{-/-} than in WT mice." But the representative images do not support that. Please quantify the intensity of IL1 β in overall and in Iba-1+ cells.

We thank the reviewer for this comment. We have quantified the overall IL-1 β positive-stained area in the retinal areas next to the CNV lesions from WT and *Tpc2*^{-/-} retina and found a reduced IL-1 β signal in *Tpc2*^{-/-} mice. The data are now included in Supplementary Fig. S7 (panel S7C). Following the reviewer's suggestion, we in addition have quantified IL-1 β intensities in Iba-1+ cells (see also response to reviewer #2) and found that the level of IL-1 β was increased in *Tpc2*^{-/-} cells, which is in line with our findings in cultured macrophages and microglia (Fig. 8). The novel data and quantification is now shown in novel Fig. 6 (panels 6B and C).

References

1. Andrei, C., P. Margiocco, A. Poggi, L.V. Lotti, M. Torrisi, and A. Rubartelli, *Phospholipases C and A2 control lysosome-mediated IL-1 β secretion: implications for inflammatory processes*. Proceedings of the National Academy of Sciences, 2004. **101**(26): p. 9745-9750.
2. Gudipaty, L., J. Munetz, P.A. Verhoef, and G.R. Dubyak, *Essential role for Ca²⁺ in regulation of IL-1 β secretion by P2X7 nucleotide receptor in monocytes, macrophages, and HEK-293 cells*. American Journal of Physiology-Cell Physiology, 2003. **285**(2): p. C286-C299.

3. Tseng, H.H.L., C.T. Vong, Y.W. Kwan, S.M.-Y. Lee, and M.P.M. Hoi, *Lysosomal Ca²⁺ signaling regulates high glucose-mediated interleukin-1 β secretion via transcription factor EB in human monocytic cells*. *Frontiers in immunology*, 2017. **8**: p. 1161.
4. Krause, T.A., A.F. Alex, D.R. Engel, C. Kurts, and N. Eter, *VEGF-production by CCR2-dependent macrophages contributes to laser-induced choroidal neovascularization*. *PLoS one*, 2014. **9**(4): p. e94313.
5. Favia, A., M. Desideri, G. Gambarà, A. D'Alessio, M. Ruas, B. Esposito, D. Del Bufalo, J. Parrington, E. Ziparo, and F. Palombi, *VEGF-induced neovascularization is mediated by NAADP and two-pore channel-2-dependent Ca²⁺ signaling*. *Proceedings of the National Academy of Sciences*, 2014. **111**(44): p. E4706-E4715.
6. Kvanta, A., *Expression and regulation of vascular endothelial growth factor in choroidal fibroblasts*. *Current eye research*, 1995. **14**(11): p. 1015-1020.
7. Shao, Z., M. Friedlander, C.G. Hurst, Z. Cui, D.T. Pei, L.P. Evans, A.M. Juan, H. Tahir, F. Duhamel, and J. Chen, *Choroid sprouting assay: an ex vivo model of microvascular angiogenesis*. *PLoS one*, 2013. **8**(7): p. e69552.
8. Wu, W.-K., O.P. Llewellyn, D.O. Bates, L.B. Nicholson, and A.D. Dick, *IL-10 regulation of macrophage VEGF production is dependent on macrophage polarisation and hypoxia*. *Immunobiology*, 2010. **215**(9-10): p. 796-803.
9. Herzog, J., S.M. Ehrlich, L. Pfitzer, J. Liebl, T. Fröhlich, G.J. Arnold, W. Mikulits, C. Haider, A.M. Vollmar, and S. Zahler, *Cyclin-dependent kinase 5 stabilizes hypoxia-inducible factor-1 α : a novel approach for inhibiting angiogenesis in hepatocellular carcinoma*. *Oncotarget*, 2016. **7**(19): p. 27108.
10. Balsler, C., A. Wolf, M. Herb, and T. Langmann, *Co-inhibition of PGF and VEGF blocks their expression in mononuclear phagocytes and limits neovascularization and leakage in the murine retina*. *Journal of neuroinflammation*, 2019. **16**(1): p. 1-12.
11. Wigg, J.P., H. Zhang, and D. Yang, *A quantitative and standardized method for the evaluation of choroidal neovascularization using MICRON III fluorescein angiograms in rats*. *PLoS One*, 2015. **10**(5): p. e0128418.
12. Gong, Y., J. Li, Y. Sun, Z. Fu, C.-H. Liu, L. Evans, K. Tian, N. Saba, T. Fredrick, and P. Morss, *Optimization of an image-guided laser-induced choroidal neovascularization model in mice*. *PLoS one*, 2015. **10**(7): p. e0132643.

June 8, 2021

RE: Life Science Alliance Manuscript #LSA-2021-01047-TR

Prof. Stylianos Michalakis
Ludwig-Maximilians-Universität München
Ophthalmology
Mathildenstr. 8
Munich, Bavaria 80336
Germany

Dear Dr. Michalakis,

Thank you for submitting your revised manuscript entitled "TPC2 promotes choroidal angiogenesis and inflammation in a mouse model of neovascular AMD.". We would be happy to publish your paper in Life Science Alliance pending final revisions necessary to meet our formatting guidelines. Please also address Reviewer 3's remaining comments in your revision.

- please remove yourself as the secondary Corresponding Author. You are already listed in the system as a primary corresponding Author
- please make sure the author order in your manuscript and our system match
- please consult our manuscript preparation guidelines <https://www.life-science-alliance.org/manuscript-prep> and make sure your manuscript sections are in the correct order
- please add callouts for Figures S1A-B; S4A-B to your main manuscript text

FIGURE CHECKS:

-Please add scale bars for Figures 1A, 2A, B; S1A, B; S2A, B; S3A, B. The size of the scale bar should be indicated in the corresponding Figure Legend.

A. FINAL FILES:

B. MANUSCRIPT ORGANIZATION AND FORMATTING:

Sincerely,

Reviewer #1 (Comments to the Authors (Required)):

The authors have adequately addressed my (minor) concerns and clarified my queries. The lack of effect of TRPML1 KO shows specificity for TPCs and not general lysosomal dysfunction. It might be worth mentioning this in the ms if for some reason the authors do not wish to include the nice data they provided in the rebuttal.

Reviewer #2 (Comments to the Authors (Required)):

I'm satisfied with the authors' reply and their new data. Please add the Figure numbers in the figure part.

Reviewer #3 (Comments to the Authors (Required)):

In this manuscript, the authors examined the role of TPC2 in the release of IL1 β , recruitment of inflammatory cells, and the effect on the laser-induced CNV using TPC2 knockout mice and inhibitors for TPC2. I saw authors put effort into revising this manuscript to address the reviewer's comments. However, their explanation about the function of TPC2 is confusing, mainly due to Fig. 3. Therefore, I recommend revising it for better understanding for readers.

Major comments.

1) Although the authors studied the function of TPC2 in microglia and macrophage, and most of the data follow the main topic, Fig. 3A-D is not. As the authors mentioned, choroidal tissue mainly has EC, pericyte, and smooth muscle cells. Thus, these data studied the function of TPC2 in either EC, pericyte, and smooth muscle cells. But they did not test the expression of TPC2 in these cell types, and they did not clearly explain that in the main text. As VEGF was provided in the culture media for choroidal sprouting assay, the VEGF amount in Fig. 3E is confusing. It is better to move Fig. 3E after Fig. 4E and explain that VEGF secretion from microglia and macrophage was not affected by deletion of TPC2 (include the data in the figure), but the reduced number of recruited cells affected the amount of VEGF in the tissue. Fig. 3A-D showed that even if the same amount of VEGF was provided, the downstream signaling in EC (or pericyte or smooth muscle cells) was affected in TPC2^{-/-}. That is an interesting but different story. It needs to be clearly mentioned in the main text.

2) In S4, the authors used IB4, NG2, and Vimentin for staining EC, pericyte, and SMC. But, some cells are double-positive for IB4 and NG2, IB4 and Vimentin. Thus, the staining seems not to work well to distinguish these cell types. These markers are widely used, so something was wrong with the staining.

Response to reviewers

We thank all three reviewers for taking the time to review this manuscript.

Added or modified sentences are always marked in red in the revised manuscript.

Response to Reviewer #1

We have included a comment on TRPML1 in the discussion section (lines 225-229).

Reviewer #2

Figure numbers have been added.

Reviewer #3

1) Although the authors studied the function of TPC2 in microglia and macrophage, and most of the data follow the main topic, Fig. 3A-D is not. As the authors mentioned, choroidal tissue mainly has EC, pericyte, and smooth muscle cells. Thus, these data studied the function of TPC2 in either EC, pericyte, and smooth muscle cells. But they did not test the expression of TPC2 in these cell types, and they did not clearly explain that in the main text. As VEGF was provided in the culture media for choroidal sprouting assay, the VEGF amount in Fig. 3E is confusing. It is better to move Fig. 3E after Fig. 4E and explain that VEGF secretion from microglia and macrophage was not affected by deletion of TPC2 (include the data in the figure), but the reduced number of recruited cells affected the amount of

VEGF in the tissue. Fig. 3A-D showed that even if the same amount of VEGF was provided, the downstream signaling in EC (or pericyte or smooth muscle cells) was affected in TPC2^{-/-}. That is an interesting but different story. It needs to be clearly mentioned in the main text.

We thank the reviewer for pointing this out. We agree and moved panel E of Fig. 3 to Fig. 4 (new panel Fig. 4F). As suggested, we also have included the data on VEGF secretion from cultured macrophages in the new Fig. S9. We now thoroughly discuss this point and have also added a sentence in the Discussion to clarify that we did not study TPC2-mediated downstream signaling in ECs.

2.) In S4, the authors used IB4, NG2, and Vimentin for staining EC, pericyte, and SMC. But, some cells are double-positive for IB4 and NG2, IB4 and Vimentin. Thus, the staining seems not to work well to distinguish these cell types. These markers are widely used, so something was wrong with the staining.

We agree with the reviewer that IB4 detection in our assay shows a signal that partly overlaps with NG2 and vimentin. In the original publication of the choroid sprouting assay (Shao, Friedlander et al., 2013) it was actually shown that the tube-like growth-cones of the endothelial cells are surrounded by NG2-positive pericytes (see Fig. 2D in (Shao et al., 2013)). Nonetheless, it must be noted that matrigel poses several difficulties for immunostainings, as also acknowledged and troubleshot by the distributing companies (see http://fscimage.fishersci.com/cmsassets/downloads/segment/Scientific/pdf/BD/bd_cellculture_matrigel_faq.pdf and <https://www.corning.com/catalog/cls/documents/faqs/CLS-DL-CC-026.pdf>).

Additionally, extracellular matrix components like collagen (contained in matrigel) are known to trigger important levels of autofluorescence, mainly when excited at shorter wavelengths (Deal, Mayes et al., 2018, Hagiwara, Hattori et al., 2011, Jun, Kim et al., 2017). For all these reasons, the detection of IB4 in choroidal sprouts could be in part affected, leading to the erroneous interpretation of a higher degree of overlap with the other signals. Given that IB4 was mainly used as a control, we have therefore omitted the IB4 staining from the supplementary Fig. S4 and have rephrased the sentence referring to this figure in the main text.

References:

- Deal J, Mayes S, Browning C, Hill S, Rider P, Boudreaux C, Rich TC, Leavesley SJ (2018) Identifying molecular contributors to autofluorescence of neoplastic and normal colon sections using excitation-scanning hyperspectral imaging. *J Biomed Opt* 24: 1-11
- Hagiwara Y, Hattori K, Aoki T, Ohgushi H, Ito H (2011) Autofluorescence assessment of extracellular matrices of a cartilage-like tissue construct using a fluorescent image analyser. *J Tissue Eng Regen Med* 5: 163-8
- Jun YW, Kim HR, Reo YJ, Dai M, Ahn KH (2017) Addressing the autofluorescence issue in deep tissue imaging by two-photon microscopy: the significance of far-red emitting dyes. *Chem Sci* 8: 7696-7704
- Shao Z, Friedlander M, Hurst CG, Cui Z, Pei DT, Evans LP, Juan AM, Tahir H, Duhamel F, Chen J (2013) Choroid sprouting assay: an ex vivo model of microvascular angiogenesis. *PLoS One* 8: e69552

June 15, 2021

RE: Life Science Alliance Manuscript #LSA-2021-01047-TRR

Prof. Stylianos Michalakis
University Hospital, LMU Munich
Ophthalmology
Mathildenstr. 8
Munich, Bavaria 80336
Germany

Dear Dr. Michalakis,

Thank you for submitting your Research Article entitled "TPC2 promotes choroidal angiogenesis and inflammation in a mouse model of neovascular AMD.". It is a pleasure to let you know that your manuscript is now accepted for publication in Life Science Alliance. Congratulations on this interesting work.

DISTRIBUTION OF MATERIALS:

Again, congratulations on a very nice paper. I hope you found the review process to be constructive and are pleased with how the manuscript was handled editorially. We look forward to future exciting submissions from your lab.

Sincerely,
